# DiffuSeq: Sequence to Sequence Text Generation with Diffusion Models

**Shansan Gong**[1], **Mukai Li**[1], **Jiangtao Feng**[1], **Zhiyong Wu**[1], **Lingpeng Kong**[2]
[1]Shark-NLP, Shanghai AI Laboratory     [2]The University of Hong Kong
{gongshansan,limukai,fengjiangtao,wuzhiyong}@pjlab.org.cn
lpk@cs.hku.hk

## Abstract

Recently, diffusion models have emerged as a new paradigm for generative models. Despite the success in domains using continuous signals such as vision and audio, adapting diffusion models to natural language is under-explored due to the discrete nature of texts, especially for conditional generation. We tackle this challenge by proposing DiffuSeq: a diffusion model designed for sequence-to-sequence (Seq2Seq) text generation tasks. Upon extensive evaluation over a wide range of Seq2Seq tasks, we find DiffuSeq achieving comparable or even better performance than six established baselines, including a state-of-the-art model that is based on pre-trained language models. Apart from quality, an intriguing property of DiffuSeq is its high diversity during generation, which is desired in many Seq2Seq tasks. We further include a theoretical analysis revealing the connection between DiffuSeq and autoregressive/non-autoregressive models. Bringing together theoretical analysis and empirical evidence, we demonstrate the great potential of diffusion models in complex conditional language generation tasks. [1]

## 1 Introduction

Among existing generative models, GAN (Goodfellow et al., 2014) suffers from the instability issue (Salimans et al., 2016), subjecting to mode collapse (Metz et al., 2017); VAE (Kingma & Welling, 2014) has to rely on surrogate objectives to approximate maximum likelihood training and Flow-based models (Dinh et al., 2017) has to use specialized architectures to construct reversible transform. Diffusion models (Ho et al., 2020; Nichol & Dhariwal, 2021) have circumvented several of these limitations and emerged as a new paradigm for generative models, theoretically underpinned by non-equilibrium thermodynamics (Sohl-Dickstein et al., 2015) and score-matching network (Song & Ermon, 2019). To date, the major breakthroughs are in domains using continuous signals, such as vision (Saharia et al., 2022a;b; Ramesh et al., 2022) and audio (Kong et al., 2020). However, extending continuous diffusion models to natural language remains an open challenge due to the inherently discrete nature of texts.

On the basis of unconditional generation in continuous space which is illustrated in Figure 1(a), existing efforts (Hoogeboom et al., 2021; Austin et al., 2021) start customizing diffusion models to text in discrete space on unconditional language modeling (i.e., free text generation). Diffusion-LM (Li et al., 2022), as in Figure 1(b), models texts in continuous space and proposes to use an extra-trained classifier as guidance (i.e., the condition signal $\mathbf{x}$) to impose subtle changes (usually complex, fine-grained constraints) on generated sentences. Nonetheless, these models do not naturally generalize to conditional language modeling (i.e., the model assigns probabilities $p(\mathbf{w}|\mathbf{x})$ to sequences of words $\mathbf{w}$ given $\mathbf{x}$). In the more general sequence-to-sequence (Seq2Seq) setting where the condition $\mathbf{x}$ is also a sequence of words, applying Diffusion-LM can be difficult. The reason is that classifiers are attributes-oriented, and we can not train hundreds-of-thousands classifiers to model the semantic meaning between conditions and generated sentences.

Seq2Seq is an essential setting in NLP that covers a wide range of important tasks such as open-ended sentence generation, dialogue, paraphrasing, and text style transfer. In this paper, we propose

---

[1]Code is available at https://github.com/Shark-NLP/DiffuSeq

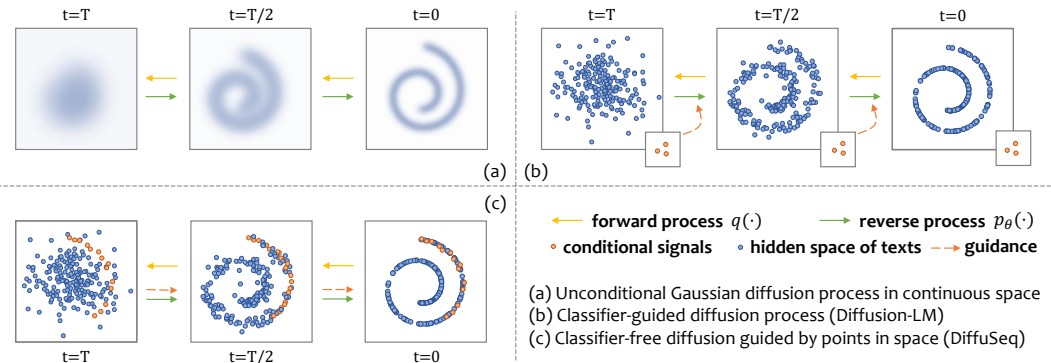

Figure 1: The demonstration of unconditional, classifier-guided, and classifier-free diffusion models.

DIFFUSEQ, depicted in Figure 1(c), a classifier-free diffusion model that supports SEQ2SEQ text generation tasks. By modeling the conditional probability of the target sentence **w** given context **x** using one single model, one advantage of DIFFUSEQ is that this paradigm allows a complete model to fit data distribution and utilize conditional guidance, rather than depending on a separate classifier.

Different from canonical generation approaches in an autoregressive (AR) left-to-right manner (Radford et al., 2019), DIFFUSEQ generates text tokens parallelly in the non-autoregressive (NAR) way. To corroborate the effectiveness of our DIFFUSEQ, we conduct experiments on four SEQ2SEQ tasks. Compared to AR and NAR models, which suffer from the "degeneration" problem (Holtzman et al., 2019) and rely on decoding strategies, DIFFUSEQ can achieve considerable sentence-level diversity without sacrificing the quality (see § 4.2).

To sum up, we make a series of technical and conceptual contributions: (a) we are the first to deploy the diffusion model on SEQ2SEQ text generation, and our proposed DIFFUSEQ as a conditional language model is trained end-to-end in a classifier-free manner; (b) we establish a theoretical connection among AR, NAR and DIFFUSEQ models, and justify DIFFUSEQ as an extension of iterative-NAR models; (c) with strong empirical evidence, we demonstrate the great potential of diffusion models in complex conditional language generation tasks.

## 2 PRELIMINARY AND PROBLEM STATEMENT

**Preliminary.** A diffusion model typically contains forward and reverse processes. Given a data point sampled from a real-world data distribution $\mathbf{z}_0 \sim q(\mathbf{z})$, the forward process gradually corrupts $\mathbf{z}_0$ into a standard Gaussian noise $\mathbf{z}_T \sim \mathcal{N}(0, \mathbf{I})$. For each forward step $t \in [1, 2, ..., T]$, the perturbation is controlled by $q(\mathbf{z}_t|\mathbf{z}_{t-1}) = \mathcal{N}(\mathbf{z}_t; \sqrt{1 - \beta_t}\mathbf{z}_{t-1}, \beta_t \mathbf{I})$, with $\beta_t \in (0, 1)$ as different variance scales. Once the forward process is completed, the reverse denoising process tries to gradually reconstruct the original data $\mathbf{z}_0$ via sampling from $\mathbf{z}_T$ by learning a diffusion model $f_\theta$.

**Problem Statement.** Many recent efforts have been devoted to adapting diffusion models to discrete texts (See § 5). However, they all focus on unconditional sequence modeling. In this paper, we target the sequence-to-sequence text generation tasks. In particular, given a $m$-length source sequence $\mathbf{w}^x = \{w_1^x, ..., w_m^x\}$, we aim to learn a diffusion model that can produce a $n$-length target sequence $\mathbf{w}^y = \{w_1^y, ..., w_n^y\}$ conditioning on the source sequence.

## 3 DIFFUSEQ

We propose DIFFUSEQ to extend vanilla diffusion models to learn conditional text generation (as shown in Figure 2), concerning the model architecture and the training objective.

**Forward Process with Partial Noising.** In the beginning of forward process, we follow Diffusion-LM (Li et al., 2022) to design an embedding function EMB(**w**) to map the discrete text **w** into a continuous space. In particular, given a pair of sequence $\mathbf{w}^x$ and $\mathbf{w}^y$, DIFFUSEQ learns a unified

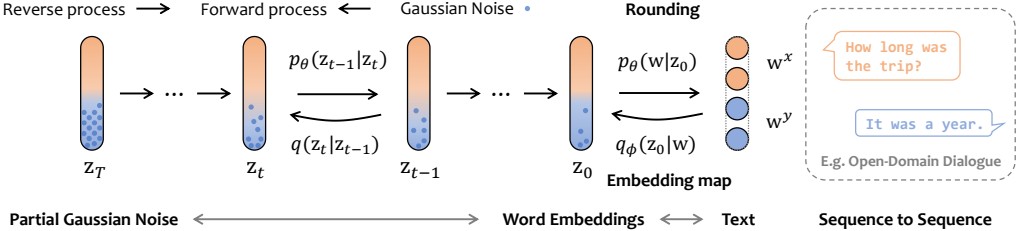

Figure 2: The diffusion process of our conditional diffusion language model DIFFUSEQ. Given the source $\mathbf{w}^x$ and the target $\mathbf{w}^y$, we pair-wisely transform them into continuous space $\mathbf{z}_0$. The partial Gaussian noise is iteratively added on the target space of $\mathbf{z}_t$.

feature space of $\mathbf{w}^x$ and $\mathbf{w}^y$ by embedding transformation and concatenation as $\text{EMB}(\mathbf{w}^{x\oplus y}) = [\text{EMB}(w_1^x), ..., \text{EMB}(w_m^x), \text{EMB}(w_1^y), ..., \text{EMB}(w_n^y)] \in \mathbb{R}^{(m+n)\times d}$. The transformation allows us to adapt discrete textual input into the standard forward process, by extending the original forward chain to a new Markov transition $q_\phi(\mathbf{z}_0|\mathbf{w}^{x\oplus y}) = \mathcal{N}(\text{EMB}(\mathbf{w}^{x\oplus y}), \beta_0 \mathbf{I})$.

We denote $\mathbf{z}_t = \mathbf{x}_t \oplus \mathbf{y}_t$ to simplify the wordings, where $\mathbf{x}_t$ and $\mathbf{y}_t$ represent parts of $\mathbf{z}_t$ that belong to $\mathbf{w}^x$ and $\mathbf{w}^y$, respectively. For each forward step $q(\mathbf{z}_t|\mathbf{z}_{t-1})$, we gradually inject noise into last step's hidden state $\mathbf{z}_{t-1}$ to obtain $\mathbf{z}_t$. Unlike conventional diffusion models that corrupt the whole $\mathbf{z}_t$ (both $\mathbf{x}_t$ and $\mathbf{y}_t$) without distinction, we only impose noising on $\mathbf{y}_t$. This modification (termed **partial noising**) allows us to adapt diffusion models for conditional language modeling.

**Reverse Process with Conditional Denoising.** The ultimate goal of the reverse process is to recover the original $\mathbf{z}_0$ by denoising $\mathbf{z}_t$: $p_\theta(\mathbf{z}_{0:T}) := p(\mathbf{z}_T) \prod_{t=1}^{T} p_\theta(\mathbf{z}_{t-1}|\mathbf{z}_t)$. We model the learning process $p_\theta(\mathbf{z}_{t-1}|\mathbf{z}_t) = \mathcal{N}(\mathbf{z}_{t-1}; \mu_\theta(\mathbf{z}_t, t), \sigma_\theta(\mathbf{z}_t, t))$ using the proposed diffusion model DIFFUSEQ: $f_\theta(\mathbf{z}_t, t)$, where the $\mu_\theta(\cdot)$ and $\sigma_\theta(\cdot)$ is the parameterization of the predicted mean and standard deviation of $q(\mathbf{z}_{t-1}|\mathbf{z}_t)$ in forward process, derived using Bayes' rule. The detailed derivations are in Appendix A. With the partial nosing strategy adopted in the forward process, we can impose the input as the condition when denoising as shown in Figure 1. The proposed conditional denoising is classifier-free by nature: we do not require extra-trained classifiers to control the denoising process.

Specifically, we use a transformer architecture to model $f_\theta$, which spontaneously models the semantic relation between $\mathbf{x}_t$ and $\mathbf{y}_t$. We compute the variational lower bound ($\mathcal{L}_{\text{VLB}}$) following the original diffusion process. $\mathcal{L}_{round}$ corresponds to rounding operation in Figure 2.

$$
\mathcal{L}_{\text{VLB}} = \mathbb{E}_{q(\mathbf{z}_{1:T}|\mathbf{z}_0)} \left[ \log \underbrace{\frac{q(\mathbf{z}_T|\mathbf{z}_0)}{p_\theta(\mathbf{z}_T)}}_{\mathcal{L}_T} + \sum_{t=2}^{T} \log \underbrace{\frac{q(\mathbf{z}_{t-1}|\mathbf{z}_0, \mathbf{z}_t)}{p_\theta(\mathbf{z}_{t-1}|\mathbf{z}_t)}}_{\mathcal{L}_{t-1}} \right.
$$
$$
\left. + \underbrace{\log \frac{q_\phi(\mathbf{z}_0|\mathbf{w}^{x\oplus y})}{p_\theta(\mathbf{z}_0|\mathbf{z}_1)}}_{\mathcal{L}_0} - \underbrace{\log p_\theta(\mathbf{w}^{x\oplus y}|\mathbf{z}_0)}_{\mathcal{L}_{\text{round}}} \right].
\tag{1}
$$

We further simplify the training objective as follows (details in Appendix A):

$$
\min_\theta \mathcal{L}_{\text{VLB}} = \min_\theta \left[ \sum_{t=2}^{T} ||\mathbf{z}_0 - f_\theta(\mathbf{z}_t, t)||^2 + ||\text{EMB}(\mathbf{w}^{x\oplus y}) - f_\theta(\mathbf{z}_1, 1)||^2 - \log p_\theta(\mathbf{w}^{x\oplus y}|\mathbf{z}_0) \right]
$$
$$
\rightarrow \min_\theta \left[ \sum_{t=2}^{T} ||\mathbf{y}_0 - \tilde{f}_\theta(\mathbf{z}_t, t)||^2 + ||\text{EMB}(\mathbf{w}^y) - \tilde{f}_\theta(\mathbf{z}_1, 1)||^2 + \mathcal{R}(||\mathbf{z}_0||^2) \right],
\tag{2}
$$

here we use $\tilde{f}_\theta(\mathbf{z}_t, t)$ to denote the fractions of recovered $\mathbf{z}_0$ corresponding to $\mathbf{y}_0$. Note that although in the first term, we only compute the loss w.r.t $\mathbf{y}_0$, due to the attention mechanism in the transformer, the reconstruction of $\mathbf{y}_0$ also takes $\mathbf{x}_0$ into account, thus the gradients from the first term

will also affect the learning of $\mathbf{x}_0$. The mathematically equivalent regularization term $\mathcal{R}(||\mathbf{z}_0||^2))$ regularize the embedding learning. We further share the embedding function between source and target sequences, enabling the training of two different feature spaces jointly. This sets DIFFUSEQ away from existing solutions in vision such as GLIDE (Nichol et al., 2022).

**Training and Inference Methods.** In our preliminary experiments, we find that the high diversity in NLP datasets and long diffusion steps often result in insufficient training. We hypothesize the reason is that sampling step $t$ uniformly causes unnecessary noise in the $\mathcal{L}_{\text{VLB}}$ objective. We hence employ importance sampling (Nichol & Dhariwal, 2021) to address this problem.

$$\mathcal{L}_{\text{VLB}} = \mathbb{E}_{t \sim p_t}\left[\frac{\mathcal{L}_t}{p_t}\right], \; p_t \propto \sqrt{\mathbb{E}[\mathcal{L}_t^2]}, \; \sum_{t=0}^{T-1} p_t = 1. \tag{3}$$

Intuitively, the importance-weighted sampling algorithm will spend more steps on diffusion steps with larger $\mathcal{L}_t$, and vice versa.

To conduct SEQ2SEQ generation given the condition $\text{EMB}(\mathbf{w}^x)$, we randomly sample $\mathbf{y}_T \sim \mathcal{N}(0, I)$ and concatenate $\mathbf{y}_T$ with $\text{EMB}(\mathbf{w}^x)$ to obtain $\mathbf{z}_T$. We can now repeat the reverse process until we arrive at $\mathbf{z}_0$. At each sampling step, an anchoring function is executed towards reparameterized $\mathbf{z}_t$. Specifically, the anchoring function: (a) operates rounding on $\mathbf{z}_t$ to map it back to word embedding space following Li et al. (2022); (b) replaces the part of recovered $\mathbf{z}_{t-1}$ that belongs to $\mathbf{w}^x$ with the original $\mathbf{x}_0$, considering that this part is recovered from corrupted $\mathbf{z}_t$ via $f_\theta$ and not strictly equals to $\mathbf{x}_0$. Note that (b) is designed for DIFFUSEQ.

To improve the quality of generation, we apply the widely used Minimum Bayes Risk (MBR) decoding strategy (Koehn, 2004). We first generate a set of candidate samples $\mathcal{S}$ from different random seeds of DIFFUSEQ and select the best output sequence that achieves the minimum expected risk under a meaningful loss function (e.g. BLEU or other cheaper metrics like precision). In practice, we use the negative BLEU score in our implementation.

**Connections to AR, Iter-NAR, and Fully-NAR Models.** To better understand the behavior of DIFFUSEQ, we give the theoretical connection to autoregressive (AR), iterative non-autoregressive (iter-NAR), and fully non-autoregressive (fully-NAR) models. We argue that DIFFUSEQ can be seen as an extension of iter-NAR model. Detailed graphical learning discrepancies of these four cases are discussed in Appendix B for reference.

AR models learn $p(\mathbf{w}_{1:n}^y | \mathbf{w}^x)$ by autoregressive decomposition based on left-context:

$$p_{\text{AR}}(\mathbf{w}_{1:n}^y | \mathbf{w}^x) = \underbrace{p(w_1^y | \mathbf{w}^x)}_{\text{initial prediction}} \underbrace{\prod_{i=1,\ldots,n-1} p(w_{i+1}^y | \mathbf{w}_{1:i}^y, \mathbf{w}^x)}_{\text{progressive left-context prediction}}, \tag{4}$$

while fully-NAR models (Gu et al., 2018; Qian et al., 2021) learn the conditional probability given independent assumption for fast inference:

$$p_{\text{fully-NAR}}(\mathbf{w}_{1:n}^y | \mathbf{w}^x) = \prod_{i=1,\ldots,n} p(w_i^y | \mathbf{w}^x). \tag{5}$$

To make a better analogy to AR and NAR models, we use a lossless way to formulate iterative NAR models (Gu et al., 2019; Ghazvininejad et al., 2019) by introducing a series of intermediate sequences $\mathbf{w}_{1:K-1}^y, \mathbf{w}_K^y = \mathbf{w}^y$ with $K$ editable iterations:

$$p_{\text{iter-NAR}}(\mathbf{w}_{1:n}^y | \mathbf{w}^x) = \sum_{\mathbf{w}_1^y,\ldots,\mathbf{w}_{K-1}^y} \underbrace{\prod_{i=1\ldots n} p(w_{1,i}^y | \mathbf{w}^x)}_{\text{initial prediction}} \underbrace{\prod_{k=1..K-1} \prod_{i=1\ldots n} p(w_{k+1,i}^y | \mathbf{w}_{k,1:n}^y, \mathbf{w}^x)}_{\text{progressive full-context prediction}}. \tag{6}$$

Previous study (Huang et al., 2022) shows that there is a gap called *conditional total correlation* between AR Eq. (4) and fully-NAR Eq. (5) learning paradigms, because of lossy decomposition of NAR models. However, when comparing iter-NAR Eq. (6) with AR Eq. (4) models, they both can be factorized into an initial prediction term and a progressive prediction process based on different context (i.e. left-context in AR and full-context in iter-NAR), and the discrepancy pointed out by

Huang et al. (2022) is therefore closed in iter-NAR assuming sufficient steps. By showing DIF-FUSEQ is an extension of the iter-NAR model, we offer a justification that it will not suffer from the conditional total correlation for the same reason.

A straight-forward way to formulate pure continuous diffusion models is to introduce a series of Gaussian noise-corrupted features along with diffusion steps: $\mathbf{y}_{1:T-1}, \mathbf{y}_0 = \mathbf{y}, \mathbf{y}_T \sim \mathcal{N}(0, \mathbf{I})$.

$$p_{\text{diffusion}}(\mathbf{w}^y|\mathbf{w}^x) = \int_{\mathbf{y}_T,\ldots,\mathbf{y}_0} \underbrace{p(\mathbf{w}^y|\mathbf{y}_0, \mathbf{w}^x)}_{\text{final prediction}} \underbrace{\prod_{t=T,\ldots,1} p(\mathbf{y}_{t-1}|\mathbf{y}_t, \mathbf{w}^x)}_{\text{progressive full-context diffusion}}, \tag{7}$$

where $p(\mathbf{y}_{t-1}|\mathbf{y}_t, \mathbf{w}^x)$ describes the diffusion step on continuous representations $\mathbf{y}$. The rounding operation in DIFFUSEQ maps the continuous vectors $\mathbf{y}$ to discrete $\mathbf{w}^y$ for each time step $t$, we in addition introduce this into Eq. (7):

$$p_{\text{DIFFUSEQ}}(\mathbf{w}^y|\mathbf{w}^x) = \sum_{\mathbf{w}_T^y,\ldots,\mathbf{w}_1^y} \int_{\mathbf{y}_T,\ldots,\mathbf{y}_0} p(\mathbf{w}^y|\mathbf{y}_0, \mathbf{w}^x) \prod_{t=T,\ldots,1} p(\mathbf{w}_t^y|\mathbf{y}_t, \mathbf{w}^x) p(\mathbf{y}_{t-1}|\mathbf{w}_t^y) \tag{8}$$

$$= \sum_{\mathbf{w}_T^y,\ldots,\mathbf{w}_1^y} \int_{\mathbf{y}_T,\ldots,\mathbf{y}_0} p(\mathbf{w}_T^y|\mathbf{y}_T, \mathbf{w}^x) \prod_{t=T-1,\ldots,0} p(\mathbf{y}_t|\mathbf{w}_{t+1}^y) p(\mathbf{w}_t^y|\mathbf{y}_t, \mathbf{w}^x). \tag{9}$$

By rearranging Eq. (8) into Eq. (9), we can see DIFFUSEQ can be seen as a more generalized form of iter-NAR Eq. (6) before marginalizing out $\{\mathbf{y}_T, \ldots, \mathbf{y}_0\}$, despite the different initialization of $\mathbf{y}_T$ [2]. A more detailed derivation is shown in Appendix C.

## 4 EXPERIMENTS

We conduct experiments to validate the effectiveness of DIFFUSEQ on four different tasks, against six strong AR/NAR baselines.

### 4.1 EXPERIMENTAL SETUP

**Tasks and Datasets.** SEQ2SEQ generation covers a wide range of tasks, among which we choose four typical and popular tasks. **Open domain dialogue** requires models to generate informative responses given a dialogue context. We use Commonsense Conversation Dataset (Zhou et al., 2018), which is extracted from Reddit single-round dialogs, with over 3 million conversational pairs. **Question generation**(QG) aims to generate questions given a context as input. To obtain sufficient training samples, we use the dataset Quasar-T (Dhingra et al., 2017) preprocessed by Lin et al. (2018), and then generate document-question pairs to obtain 119K training samples (details in Appendix D.1). **Text simplification** aims to revise the complex text into sequences with simplified grammar and word choice. Jiang et al. (2020) constructs a corpus consisting of 677K complex-simple sentences with revision alignment. **Paraphrase** task generates an alternative surface form in the same language expressing the same semantic content. We adopt widely used QQP [3] sourced from the community question answering forum Quora, with 147K positive pairs.

**Baselines.** We consider three groups of models as baselines, covering both AR and NAR architectures. The first group of methods adopts encoder-decoder architecture (Cho et al., 2014) which is well-studied for SEQ2SEQ tasks, and we conduct experiments on two popular models: GRU with attention and Transformer (Vaswani et al., 2017). The second group is the finetuned large pre-trained language model (PLM), among which GPT2 (Radford et al., 2019) has demonstrated great success in almost all SEQ2SEQ tasks. We further compare to GPVAE (Du et al., 2022), which augments a pre-trained T5 (Raffel et al., 2020) with VAE to improve the generation diversity. For the last group of baselines, we consider LevT (Gu et al., 2019), a widely used, strong iterative NAR model. All baselines are trained following instructions in their papers, and details can be found in Appendix D.2.

---

[2] For NAR models, $\mathbf{y}_T$ is uniform copied from the source sentence or *unk*'s token embedding (Gu et al., 2018); for diffusion models, $\mathbf{y}_T$ is sampled from normal distribution $\mathcal{N}(0, \mathbf{I})$.

[3] https://www.kaggle.com/c/quora-question-pairs

Table 1: The overall results of different methods on different SEQ2SEQ tasks. The first group ◇ of methods adopt autoregressive encoder-decoder architecture and the second group ● is the fine-tuned large pre-trained language model (also in autoregressive manner) while the last group ‡ is non-autoregressive. The best results are **bold**, and the best results without PLMs are underlined.

| Tasks | Methods | BLEU↑ | R-L↑ | Score↑ | dist-1↑ | selfB↓ / div-4↑ | Len |
|---|---|---|---|---|---|---|---|
| Open Domain Dialogue | GRU-attention ◇ | 0.0068 | 0.1054 | 0.4128 | 0.8998 | 0.8008/0.1824 | 4.46 |
| | Transformer-base ◇ | **0.0189** | 0.1039 | 0.4781 | 0.7493 | 0.3698/0.6472 | 19.5 |
| | GPT2-base FT● | 0.0108 | **0.1508** | 0.5279 | 0.9194 | 0.0182/0.9919 | 16.8 |
| | GPT2-large FT ● | 0.0125 | 0.1002 | **0.5293** | 0.9244 | 0.0213/0.9938 | 16.8 |
| | GPVAE-T5● | 0.0110 | 0.1009 | 0.4317 | 0.5625 | 0.3560/0.5551 | 20.1 |
| | NAR-LevT ‡ | 0.0158 | 0.0550 | 0.4760 | **0.9726** | 0.7103/0.1416 | 4.11 |
| | DIFFUSEQ (Ours) ‡ | 0.0139 | 0.1056 | 0.5131 | 0.9467 | **0.0144/0.9971** | 13.6 |
| Question Generation | GRU-attention ◇ | 0.0651 | 0.2617 | 0.5222 | 0.7930 | 0.9999/0.3178 | 10.1 |
| | Transformer-base ◇ | 0.1663 | 0.3441 | 0.6307 | 0.9309 | 0.3265/0.7720 | 10.3 |
| | GPT2-base FT ● | 0.0741 | 0.2714 | 0.6052 | 0.9602 | **0.1403/0.9216** | 10.0 |
| | GPT2-large FT ● | 0.1110 | 0.3215 | **0.6346** | **0.9670** | 0.2910/0.8062 | 9.96 |
| | GPVAE-T5● | 0.1251 | 0.3390 | 0.6308 | 0.9381 | 0.3567/0.7282 | 11.4 |
| | NAR-LevT ‡ | 0.0930 | 0.2893 | 0.5491 | 0.8914 | 0.9830/0.4776 | 6.93 |
| | DIFFUSEQ (Ours)‡ | **0.1731** | **0.3665** | 0.6123 | 0.9056 | 0.2789/0.8103 | 11.5 |
| Text Simplification | GRU-attention ◇ | 0.3256 | 0.5602 | 0.7871 | 0.8883 | 0.9998/0.3313 | 18.9 |
| | Transformer-base ◇ | 0.2693 | 0.4907 | 0.7381 | 0.8886 | 0.6924/0.5095 | 18.5 |
| | GPT2-base FT ● | 0.3083 | 0.5461 | 0.8021 | 0.9439 | 0.5444/0.6047 | 16.1 |
| | GPT2-large FT ● | 0.2693 | 0.5111 | 0.7882 | 0.9464 | 0.6042/0.5876 | 15.4 |
| | GPVAE-T5 ● | 0.3392 | 0.5828 | **0.8166** | 0.9308 | 0.8147/0.4355 | 18.5 |
| | NAR-LevT ‡ | 0.2052 | 0.4402 | 0.7254 | 0.9715 | 0.9907/0.3271 | 8.31 |
| | DIFFUSEQ (Ours) ‡ | **0.3622** | **0.5849** | 0.8126 | 0.9264 | **0.4642/0.6604** | 17.7 |
| Paraphrase | GRU-attention ◇ | 0.1894 | 0.5129 | 0.7763 | 0.9423 | 0.9958/0.3287 | 8.30 |
| | Transformer-base ◇ | **0.2722** | 0.5748 | 0.8381 | 0.9748 | 0.4483/0.7345 | 11.2 |
| | GPT2-base FT ● | 0.1980 | 0.5212 | 0.8246 | 0.9798 | 0.5480/0.6245 | 9.67 |
| | GPT2-large FT ● | 0.2059 | 0.5415 | 0.8363 | **0.9819** | 0.7325/0.5020 | 9.53 |
| | GPVAE-T5 ● | 0.2409 | **0.5886** | **0.8466** | 0.9688 | 0.5604/0.6169 | 9.60 |
| | NAR-LevT ‡ | 0.2268 | 0.5795 | 0.8344 | 0.9790 | 0.9995/0.3329 | 8.85 |
| | DIFFUSEQ (Ours) ‡ | 0.2413 | 0.5880 | 0.8365 | 0.9807 | **0.2732/0.8641** | 11.2 |

**Evaluation.** We evaluate the generated sequences from two aspects: quality and diversity. To evaluate the quality, we use the standard metric BLEU (Papineni et al., 2002) and ROUGE (Lin, 2004) score. Since string-similarity-based metrics can be unsatisfactory for open-ended generation, we also report BERTScore (Zhang et al., 2019) that assesses the semantic similarity between generated sentences and references. Details are in Appendix D.4. Higher scores of BLEU, ROUGE and BERTScore reflect better performance. As for diversity, we use distinct unigram (dist-1) to measure intra-diversity within each generated sentence, where the lower dist-1 indicates that the generated sentence contains more repeated words. For sentence-level diversity evaluation, we consider sentence-level self-BLEU (Zhu et al., 2018) to measure the n-gram overlap between the set of outputs w.r.t one source sentence, and we additionally use diverse 4-gram (div-4) (Deshpande et al., 2019) to measure the ratio of distinct 4-grams in the set of outputs per source sentence. The lower self-BLEU and higher div-4 suggest higher diversity of generation. For each method including DIFFUSEQ, we generate 3 samples for each source sentence to compute the diversity metrics.

**Implementation Details.** Our DIFFUSEQ is based on the 12 layers of Transformer with 12 attention heads, where the time step embedding is plugged akin to the position embedding. The maximum sequence length is 128, with embedding dimension $d = 128$, diffusion steps $T = 2,000$ and a square-root noise schedule. To reduce the out-of-vocabulary generation, we apply Byte Pair Encoding (Sennrich et al., 2016) to construct the vocabulary. After conducting the diversity beam

Table 2: Sample outputs in QQP test set, conditioned on the same **x**.

| *Original sentence*: *How do I make friends.* | | *Paraphrase reference*: *How to make friends ?* |
|---|---|---|
| **GPT2-large finetune** | **GPVAE-T5** | **DIFFUSEQ** |
| How can I make friends? | How can I make friends? | How can I make friends better? |
| How can I make friends? | How do I make friends? | How can I make friends? |
| How can I make friends? | How can I make friends? | How do you make friends? |
| How can I make friends? | How can I make friends? | What is the best way to make friends? |
| How do I make friends and keep them? | What's the best way to make friends and make make friends? | How can I make friends and more something? |

search (DBS) (Vijayakumar et al., 2016) for the Transformer-base model and GPT model, we find that DBS does not always promote diversity over temperature sampling and therefore we list the best diversity results. We compute the accuracy metrics of DIFFUSEQ using MBR with the size of candidate samples $|\mathcal{S}| = 10$. The experiment is deployed on NVIDIA A100 Tensor Core GPUs, and we use 4 GPUs on training and single GPU on sampling.

## 4.2 MAIN RESULTS

As shown in Table 1, we conclude that DIFFUSEQ achieves comparable or even higher generation quality compared with strong baselines. At the same time, DIFFUSEQ consistently demonstrates its superiority in generating diverse outputs given the same input sequence.

As we can see from Table 1, DIFFUSEQ wins competitions over at least one quality metric against 6 baselines × 4 tasks. Although NAR models such as LevT can also outperform AR baselines sometimes, they still lag well behind DIFFUSEQ by large margins (i.e., relative improvements over 50% for BLEU in QG task and R-L in Dialogue task). Even compared with pre-trained then finetuned GPT2 models, DIFFUSEQ still delivers superior performance than the base variant, and is comparable with the large variant, which has 8.2 times more parameters than DIFFUSEQ. These empirical results amply support our findings in § 3, where we theoretically analyze the potential of diffusion models in modeling text sequences compared with AR models given sufficient diffusion steps.

DIFFUSEQ, as a member of the deep generative model family, also exhibit the capacity to generate highly diverse sequences. As suggested by self-BLEU (lower is better) and div-4 (higher is better), in almost all cases, DIFFUSEQ significantly outperforms 4 AR baselines in terms of sentence-level diversity (i.e., producing diverse outputs given the same input). For diversity in word choice within one sentence, we consider dist-1: a higher dist-1 indicates less repetition within a sentence. As we can see from Table 1, DIFFUSEQ has less repetition compared with encoder-decoder methods, but still fall behind the pre-trained GPT2 models (the same situation with BERTScore). These results suggest there is still room for improvement (e.g., use pre-training techniques) in diffusion models' token-level choice. Different from NAR-LevT, DIFFUSEQ does not rely on an extra length prediction module but automatically decides by the padding token instead and is able to generate longer output sentences, indicated by the last column for average generation length.

In Table 2, we provide examples to showcase DIFFUSEQ's ability to generate diverse samples. More examples can be found in Appendix D.5.

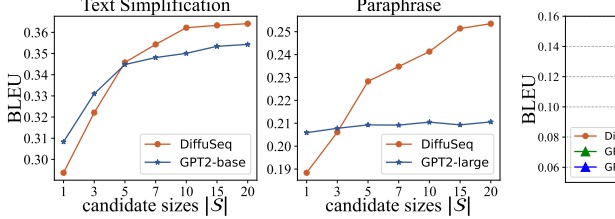
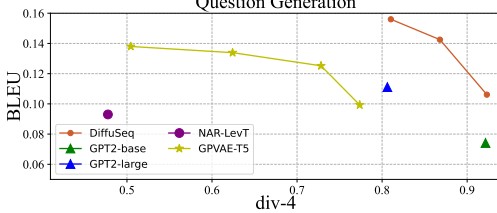

Figure 3: The increase of BLEU score with different candidate sizes $|\mathcal{S}|$.

Figure 4: Trade-off between quality and diversity (details in Appendix D.3).

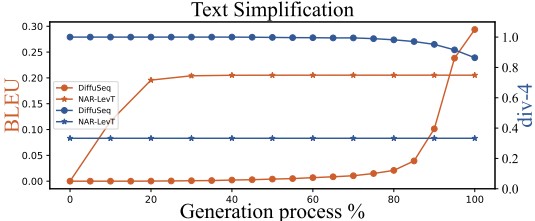 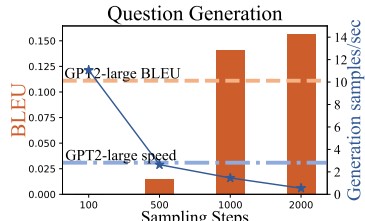

Figure 5: The curve of BLEU/div-4 score along with generation process (percentage of steps).

Figure 6: The BLEU and inference speed of DIFFUSEQ and GPT2-large.

## 4.3 ANALYSIS

We conduct a series of analysis to investigate the effectiveness of different aspects in DIFFUSEQ.

**Diversity Ensures Quality.** Generating high-quality texts with high diversity is an important requirement for many text generation applications and the trade-off between quality and diversity is always a critical concern in open-ended NLG tasks (Zhang et al., 2021). Different from AR models relying on the decoding strategy like temperature and nucleus sampling (Holtzman et al., 2019) and VAE models sampling latent variable from Gaussian Prior, the natural advantage of DIFFUSEQ is to generate different sentences along with a series of random Gaussian noise. In Figure 4, we elucidate that DIFFUSEQ have better trade-off between generation quality (BLEU) and sentence-level diversity (div-4). Here we further demonstrate that the high diversity provided by DIFFUSEQ can be turned into better quality.

MBR is a common strategy to improve generation quality by aggregating and ranking candidate sequences, and we find that the upper bound of MBR is decided by a diversified candidate set. To valid this, we simultaneously apply MBR on both DIFFUSEQ and GPT2 with various candidate sizes $|\mathcal{S}|$. The results are shown in Figure 3. As we can see, DIFFUSEQ lags behind GPT2 without using MBR ($|\mathcal{S}| = 1$) or with a small candidate set ($|\mathcal{S}| = 3$). However, as $|\mathcal{S}|$ increases, DIFFUSEQ starts to outperform GPT2 by an increasing margin. The reason is that autoregressive models like GPT2 tend to generate highly similar candidates (as discussed in § 4.2), which impedes the effectiveness of MBR. As $|\mathcal{S}|$ increases to 20, DIFFUSEQ still shows better rising trends than GPT2. Our findings also stress the importance of better ranking methods in diffusion research.

**Step-wise Analysis against Iterative NAR.** Given the underlying theoretical connection between iterative NAR and DIFFUSEQ discussed in § 3, we empirically investigate the behavior of LevT and DIFFUSEQ by analyzing their step-wise quality (i.e. BLEU) and diversity (i.e. div-4) curves. As is suggested in Figure 5, LevT grows fiercely in quality at the very beginning of generation, and quickly slows down in the successive refinement process. But DIFFUSEQ behaves differently, with BLEU score growing slowly at first, increasing rapidly as the diffusion process progresses and finally surpassing LevT. It is also observed that the diversity of both LevT and DIFFUSEQ is determined at the very early stage regardless of future refinement or diffusion, where DIFFUSEQ consistently outperforms LevT on diversity at any stage of generation. We conjecture that DIFFUSEQ explores more possible results at the first half of generation process, and soon converges to several potential candidates when it is closed to the end of steps. In this case, DIFFUSEQ shows its capacity to take both generation quality and diversity into consideration, and this is the capacity that iterative-NAR and even AR models can not obtain, due to the different learning paradigms.

**Inference Speed.** The slow sampling speed is one of the major concerns about diffusion models. Here we fix the number of diffusion steps during training for DIFFUSEQ while shrinking the inference steps following DDIM (Song et al., 2020). As we can see from Figure 6, when reducing the inference to 1,000 diffusion steps on single GPU, DIFFUSEQ achieves a higher BLEU score than GPT2-large yet registers a closer inference speed to GPT2-large.

**Effectiveness of Joint Training.** In DIFFUSEQ, the representations of $\mathbf{w}^x$ and $\mathbf{w}^y$ are jointly trained using the same embedding function EMB($\cdot$) (stated in § 3). To validate the effectiveness

Table 3: Results with or without joint training for Question Generation task.

| Setting | BLEU↑ | R-L↑ | Score↑ | selfB↓ / div-4↑ |
|---|---|---|---|---|
| DIFFUSEQ (w/o reranking) | 0.1567 | 0.3484 | 0.5947 | 0.2789/0.8103 |
| Fix EMB($\mathbf{w}^x$) as pre-trained | 0.0110 | 0.0687 | 0.3769 | 0.0174/0.9376 |

of this joint training strategy, we compared it with the training strategy commonly used in text-to-image diffusion models (Nichol et al., 2022; Ramesh et al., 2022). In particular, we decouple the training of EMB($\mathbf{w}^x$) and EMB($\mathbf{w}^y$) by replacing EMB($\mathbf{w}^x$) with representations extracted from a pre-trained BERT-tiny model (Turc et al., 2019). From Table 3, we find that the decoupled training strategy results in poor performance.

## 5 RELATED WORK

**Diffusion Models for Text Modeling.** Text-to-Image generation using diffusion models has developed many potential applications. Models such as Imagen (Saharia et al., 2022b) and DALL-E (Ramesh et al., 2022) are usually two-staged relying on the pre-trained models, requiring the alignment between the embedding vectors from two sources. GLIDE (Nichol et al., 2022) explores diffusion model with classifier-free (Ho & Salimans, 2022) guidance by setting guidance scale during training. The target space of these models is not discrete text space but stable vectors of pixel values. There are other works of diffusion on text generation, but they stick to the original encoder-decoder architecture and the diffusion process is interspersed on the decoder (Savinov et al., 2021), or the latent space (Yu et al., 2022).

For text generation using the diffusion models, Hoogeboom et al. (2021) introduce the multinomial diffusion for character-level text generation, the forward categorical noise is applied through the Markov transition matrix. Austin et al. (2021) generalize discrete text diffusion models by introducing the absorbing state ([MASK]). However, discrete diffusion models may suffer from the scaling of the one-hot row vectors, and they only generate text samples unconditionally in discrete space. Diffusion-LM (Li et al., 2022) and Analog Bits (Chen et al., 2022) propose a new language model diffused on the continuous latent representations, with different mapping functions that connect the discrete and continuous space of texts. Compared with our work, we focus on the SEQ2SEQ diffusion models for text generation in the continuous space and our work is the first to explore this setting to the best of our knowledge.

**Diffusion Models for Conditional Generation.** Related to conditional-VAE (Zhao et al., 2017), we can consider the latent encoded input $\mathbf{x}$ as a condition. Diffusion-LM (Li et al., 2022) adopts the plug-and-play approaches (Dathathri et al., 2020) to compose fine-grained constraints on the generated sentences, but it fails to condition on the whole source sentence in SEQ2SEQ tasks. Noted that this controllable generation method is orthogonal to our DIFFSEQ, in other words, we can further add classifier-guided constraints on the SEQ2SEQ output to further control the text generation. There are other conditional diffusion models on the time series prediction like CSDI (Tashiro et al., 2021) or audio generation like WaveGrad (Chen et al., 2021), but their class conditions are usually attributes that are easy to model, while the contextual texts as conditions are much more complex.

## 6 CONCLUSIONS

We propose DIFFUSEQ to tackle SEQ2SEQ tasks in a diffusion way, which contains the strong potential to achieve better generation quality and diversity trade-off. The capability enables favorable characteristics of DIFFUSEQ to further enhance the quality of final results, by leveraging a minimum Bayes risk decoding algorithm. Besides, we theoretically connect the AR and NAR models to DIFFUSEQ, and show that DIFFUSEQ is a powerful extension of iterative-NAR model. The empirical results demonstrate that DIFFUSEQ is also a powerful model for text generation, matching or even surpassing competitive AR, iterative NAR, and large-scale pre-trained models on quality and diversity. Given the limited progress of current diffusion models on text generation, our study addresses promising achievements by such a new sequence-to-sequence learning paradigm.

## ACKNOWLEDGMENTS

We would like to thank the anonymous reviewers and other peers for their valuable advice, and we also acknowledge Chenxin An's efforts to update the generation results for the Transformer-base model on QG and Paraphrasing tasks. This work is partially supported by the Shanghai Committee of Science and Technology (Grant No. 21DZ1100100) and the joint research scheme of the National Natural Science Foundation of China (NSFC) and the Research Grants Council (RGC) under grant number N_HKU714/21.

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

## A  OBJECTIVE DERIVATIONS OF DIFFUSEQ

The diffusion model is well-known as its ability to achieve the trade-off between flexibility and tractability of the models' probability distributions, compared with GAN, VAE and Flow-based models. Following Ho et al. (2020); Nichol & Dhariwal (2021); Song et al. (2020), we systematically define the forward noising process and reverse denoising process on latent continuous space $\mathbf{z}$.

The *forward* noising is to perturb the structure of data $\mathbf{z}_0$. $\mathbf{z}_0$ is finally changed into the partial Gaussian noise with $\mathbf{y}_T \sim \mathcal{N}(0, \mathbf{I})$ through $T$-step forward random disturbance

$$q(\mathbf{z}_t|\mathbf{z}_{t-1}) = \mathcal{N}(\mathbf{z}_t; \sqrt{1 - \beta_t}\mathbf{z}_{t-1}, \beta_t\mathbf{I}), \tag{10}$$

with $t = 1, 2, ..., T$ and $\{\beta_t \in (0,1)\}_{t=1}^T$ are the variance schedule. Let $\alpha_t = 1 - \beta_t$ and $\bar{\alpha}_t = \prod_{i=1}^t \alpha_i$, we have:

$$\begin{aligned}
\mathbf{z}_t &= \sqrt{\alpha_t}\mathbf{z}_{t-1} + \sqrt{1 - \alpha_t}\epsilon_{t-1} = \sqrt{\alpha_t\alpha_{t-1}}\mathbf{z}_{t-2} + \sqrt{1 - \alpha_t\alpha_{t-1}}\bar{\epsilon}_{t-2} \\
&= ... = \sqrt{\bar{\alpha}_t}\mathbf{z}_0 + \sqrt{1 - \bar{\alpha}_t}\epsilon,
\end{aligned} \tag{11}$$

where $\epsilon$ stands for Gaussian noises. In the end, $q(\mathbf{z}_t|\mathbf{z}_0) = \mathcal{N}(\mathbf{z}_t; \sqrt{\bar{\alpha}_t}\mathbf{z}_0, (1 - \bar{\alpha}_t)\mathbf{I})$. We use a sqrt noise schedule in Diffusion-LM (Li et al., 2022), that is, $\bar{\alpha}_t = 1 - \sqrt{t/T + s}$ with $s$ as a small constant at the start of noise level. The *reverse* process then denoises $\mathbf{z}_t$, aiming to recover original $\mathbf{z}_0$, and is defined as:

$$p_\theta(\mathbf{z}_{0:T}) := p(\mathbf{z}_T)\prod_{t=1}^T p_\theta(\mathbf{z}_{t-1}|\mathbf{z}_t), \quad p_\theta(\mathbf{z}_{t-1}|\mathbf{z}_t) = \mathcal{N}(\mathbf{z}_{t-1}; \mu_\theta(\mathbf{z}_t, t), \sigma_\theta(\mathbf{z}_t, t)). \tag{12}$$

The learning of $p_\theta$ is based on our diffusion model DIFFUSEQ: $f_\theta(\mathbf{z}_t, t)$, where the $\mu_\theta(\cdot)$ and $\sigma_\theta(\cdot)$ is the predicted parameterization of the mean and standard variation of $q(\mathbf{z}_t|\mathbf{z}_{t-1})$ in forward process. Using Bayes' rule:

$$q(\mathbf{z}_{t-1}|\mathbf{z}_t, \mathbf{z}_0) = q(\mathbf{z}_t|\mathbf{z}_{t-1}, \mathbf{z}_0)\frac{q(\mathbf{z}_{t-1}|\mathbf{z}_0)}{q(\mathbf{z}_t|\mathbf{z}_0)} \tag{13}$$

Substitute Eq. (11) to it and we can get the parameterized mean of $q(\mathbf{z}_{t-1}|\mathbf{z}_t, \mathbf{z}_0)$:

$$\mu_t(\mathbf{z}_t, \mathbf{z}_0) = \frac{\sqrt{\alpha_t}(1 - \bar{\alpha}_{t-1})}{1 - \bar{\alpha}_t}\mathbf{z}_t + \frac{\sqrt{\bar{\alpha}_{t-1}}\beta_t}{1 - \bar{\alpha}_t}\mathbf{z}_0, \tag{14}$$

and for brevity, we short the coefficient of $\mathbf{z}_t$ and $\mathbf{z}_0$ as $\mathcal{U}$ and $\mathcal{E}$ respectively.

We can use the variational lower bound to optimize the negative log-likelihood $\mathbb{E}[-\log p_\theta(\mathbf{x}_0)] \leq \mathcal{L}_{\text{VLB}}$. The objective can be further rewritten to be a combination of several KL-divergence and entropy terms following Sohl-Dickstein et al. (2015).

$$\begin{aligned}
\mathcal{L}_{\text{VLB}} = \mathcal{L}_T + \mathcal{L}_{T-1} + \cdots + \mathcal{L}_0 = \mathbb{E}_{q(\mathbf{z}_{1:T}|\mathbf{z}_0)} &\left[ \log\frac{q(\mathbf{z}_T|\mathbf{z}_0)}{p_\theta(\mathbf{z}_T)} + \sum_{t=2}^T \log\frac{q(\mathbf{z}_{t-1}|\mathbf{z}_0, \mathbf{z}_t)}{p_\theta(\mathbf{z}_{t-1}|\mathbf{z}_t)} \right. \\
&\left. + \log\frac{q_\phi(\mathbf{z}_0|\mathbf{w}^{x\oplus y})}{p_\theta(\mathbf{z}_0|\mathbf{z}_1)} - \log p_\theta(\mathbf{w}^{x\oplus y}|\mathbf{z}_0) \right].
\end{aligned} \tag{15}$$

For $1 \leq t \leq T - 1$, we compute the parameterization of $\mathcal{L}_t$ by substituting Eq. (14) to minimize the difference from $\mu_t$ and $\mu_\theta$ following Ho et al. (2020):

$$\begin{aligned}
\mathcal{L}_t &= \mathbb{E}_{\mathbf{z}_0}\left[\log\frac{q(\mathbf{z}_t|\mathbf{z}_0, \mathbf{z}_{t+1})}{p_\theta(\mathbf{z}_t|\mathbf{z}_{t+1})}\right] = \mathbb{E}_{\mathbf{z}_0}\left[\frac{1}{C}||\mu_t(\mathbf{z}_t, \mathbf{z}_0) - \mu_\theta(\mathbf{z}_t, t)||^2\right] \\
&= \mathbb{E}_{\mathbf{z}_0}\left[\frac{1}{C}||\mathcal{U}\mathbf{z}_t + \mathcal{E}\mathbf{z}_0 - (\mathcal{U}\mathbf{z}_t + \mathcal{E}f_\theta(\mathbf{z}_t, t))||^2\right] = \frac{\mathcal{E}}{C}\mathbb{E}_{\mathbf{z}_0}[||\mathbf{z}_0 - f_\theta(\mathbf{z}_t, t)||^2],
\end{aligned} \tag{16}$$

where $\mathcal{C} = 2||\sigma_\theta||^2$ is a loss independent constant. Then the optimization of training loss $\min_\theta \mathcal{L}_{\text{VLB}}$ can be further simplified as:

$$\min_\theta \left[ ||\mu(\mathbf{z}_T)||^2 + \sum_{t=2}^{T} ||\mathbf{z}_0 - f_\theta(\mathbf{z}_t, t)||^2 + ||\text{EMB}(\mathbf{w}^{x\oplus y}) - f_\theta(\mathbf{z}_1, 1)||^2 - \log p_\theta(\mathbf{w}^{x\oplus y}|\mathbf{z}_0) \right]$$

$$\rightarrow \min_\theta \left[ \sum_{t=2}^{T} ||\mathbf{z}_0 - f_\theta(\mathbf{z}_t, t)||^2 + ||\text{EMB}(\mathbf{w}^{x\oplus y}) - f_\theta(\mathbf{z}_1, 1)||^2 - \log p_\theta(\mathbf{w}^{x\oplus y}|\mathbf{z}_0) \right]$$

$$\rightarrow \min_\theta \left[ \sum_{t=2}^{T} ||\mathbf{y}_0 - \tilde{f}_\theta(\mathbf{z}_t, t)||^2 + ||\text{EMB}(\mathbf{w}^y) - \tilde{f}_\theta(\mathbf{z}_1, 1)||^2 + \mathcal{R}(||\mathbf{z}_0||^2) \right].$$

(17)

# B  GRAPHICAL MODELS OF AR, FULLY NAR, ITERATIVE NAR AND DIFFUSEQ MODELS

We start from the conditional sequence generation problem, which aims to learn a conditional probability $p(\mathbf{w}_{1:n}^y|\mathbf{w}^x)$ with $\mathbf{w}^x$ and $\mathbf{w}^y$. AR models learn $p(\mathbf{w}_{1:n}^y|\mathbf{w}^x)$ by autoregressive decomposition based on left-context:

$$p_{\text{AR}}(\mathbf{w}_{1:n}^y|\mathbf{w}^x) = \underbrace{p(w_1^y|\mathbf{w}^x)}_{\text{initial prediction}} \underbrace{\prod_{i=1,\ldots,n-1} p(w_{i+1}^y|\mathbf{w}_{1:i}^y, \mathbf{w}^x)}_{\text{progressive left-context prediction}},$$

(18)

consisting of an initial prediction and an autoregressive left-context prediction process, while fully-NAR models (Gu et al., 2018; Qian et al., 2021) learn the conditional probability given independent assumption for fast inference:

$$p_{\text{fully-NAR}}(\mathbf{w}_{1:n}^y|\mathbf{w}^x) = \prod_{i=1,\ldots,n} p(w_i^y|\mathbf{w}^x).$$

(19)

To make a better analogy to AR and NAR models, we use a lossless way to formulate iterative NAR models (Gu et al., 2019; Ghazvininejad et al., 2019) by introducing a series of intermediate sequences $\mathbf{w}_{1:K-1}^y, \mathbf{w}_K^y = \mathbf{w}^y$ as:

$$p_{\text{iter-NAR}}(\mathbf{w}_{1:n}^y|\mathbf{w}^x) = \sum_{\mathbf{w}_1^y,\ldots,\mathbf{w}_{K-1}^y} p(\mathbf{w}_1^y|\mathbf{w}^x) \prod_{k=1\ldots K-1} p(\mathbf{w}_{k+1}^y|\mathbf{w}_k^y, \mathbf{w}^x)$$

$$= \sum_{\mathbf{w}_1^y,\ldots,\mathbf{w}_{K-1}^y} p(\mathbf{w}_1^y|\mathbf{w}^x) \prod_{k=1\ldots K-1} p(\mathbf{w}_{k+1}^y|\mathbf{w}_k^y, \mathbf{w}^x)$$

$$= \sum_{\mathbf{w}_1^y,\ldots,\mathbf{w}_{K-1}^y} \underbrace{\prod_{i=1\ldots n} p(w_{1,i}^y|\mathbf{w}^x)}_{\text{initial prediction}} \underbrace{\prod_{k=1\ldots K-1} \prod_{i=1\ldots n} p(w_{k+1,i}^y|\mathbf{w}_{k,1:n}^y, \mathbf{w}^x)}_{\text{progressive full-context prediction}}$$

(20)

Previous study (Huang et al., 2022) shows that there is a gap called *conditional total correlation* between AR and fully-NAR learning paradigms, because of the lossy decomposition of NAR models. This gap is mainly responsible for the performance drop from AR to NAR models. However, when comparing iter-NAR, Eq. (20), with AR models, they both can be factorized into an initial prediction term and a progressive prediction process based on different context (i.e. left-context in AR and full-context in iter-NAR). The discrepancy as pointed out by Huang et al. (2022) is therefore closed in iter-NAR assuming sufficient steps. By showing DIFFUSEQ is an extension of the iter-NAR model, we offer a justification that it will not suffer from the conditional total correlation for the same reason.

A straight-forward way to formulate naive diffusion models is to introduce a series of Gaussian noise-corrupted features $\mathbf{y}_{1:T-1}, \mathbf{y}_0 = \mathbf{y}, \mathbf{y}_T \sim \mathcal{N}(0, \mathbf{I})$ on continuous space as:

$$p_{\text{diffusion}}(\mathbf{w}^y|\mathbf{w}^x) = \int_{\mathbf{y}_T,...,\mathbf{y}_0} \underbrace{p(\mathbf{w}^y|\mathbf{y}_0, \mathbf{w}^x)}_{\text{final-step prediction}} \underbrace{\prod_{t=T,...,1} p(\mathbf{y}_{t-1}|\mathbf{y}_t, \mathbf{w}^x)}_{\text{progressive full-context diffusion}} \tag{21}$$

$$= \int_{\mathbf{y}_T,...,\mathbf{y}_0} \prod_{i=1,...,n} p(\mathbf{w}_i^y|\mathbf{y}_{0,i}, \mathbf{w}^x) \prod_{t=T,...,1} \prod_{i=1,...,n} p(\mathbf{y}_{t-1,i}|\mathbf{y}_t, \mathbf{w}^x) \tag{22}$$

where $p(\mathbf{y}_{t-1}|\mathbf{y}_t, \mathbf{w}^x)$ describes the diffusion process on contiguous representations $\mathbf{y}$. The total number of diffusion steps is denoted as $T$. Thereafter we omit the independent decomposition on $\mathbf{w}^y$ and $\mathbf{y}_t$. To apply diffusion models on discrete space, the rounding operation in DIFFUSEQ maps the continuous vectors $\mathbf{y}$ to discrete $\mathbf{w}^y$ for each time step $t$, we hence in addition introduce both contiguous feature $\mathbf{y}$ and the discrete text $\mathbf{w}^y$ to represent the discrete text into Eq. (21):

$$p(\mathbf{w}^y|\mathbf{w}^x) = \sum_{\mathbf{w}_T^y,...,\mathbf{w}_1^y} \int_{\mathbf{y}_T,...,\mathbf{y}_0} p(\mathbf{w}_T^y|\mathbf{y}_T, \mathbf{w}^x) \prod_{t=T-1,...,0} p(\mathbf{w}_t^y|\mathbf{y}_t, \mathbf{w}^x) p(\mathbf{y}_t|\mathbf{w}_{t+1}^y) \tag{23}$$

$$= \sum_{\mathbf{w}_T^y,...,\mathbf{w}_1^y} \int_{\mathbf{y}_T,...,\mathbf{y}_0} p(\mathbf{w}^y|\mathbf{y}_0, \mathbf{w}^x) \prod_{t=T,...,1} p(\mathbf{y}_{t-1}|\mathbf{w}_t^y) p(\mathbf{w}_t^y|\mathbf{y}_t, \mathbf{w}^x) \tag{24}$$

$$= \int_{\mathbf{y}_T,...,\mathbf{y}_0} p(\mathbf{w}^y|\mathbf{y}_0, \mathbf{w}^x) \sum_{\mathbf{w}_T^y,...,\mathbf{w}_1^y} \prod_{t=T,...,1} p(\mathbf{y}_{t-1}|\mathbf{w}_t^y) p(\mathbf{w}_t^y|\mathbf{y}_t, \mathbf{w}^x) \tag{25}$$

$$= \int_{\mathbf{y}_T,...,\mathbf{y}_0} p(\mathbf{w}^y|\mathbf{y}_0, \mathbf{w}^x) \prod_{t=T,...,1} \sum_{\mathbf{w}_t^y} p(\mathbf{y}_{t-1}|\mathbf{w}_t^y) p(\mathbf{w}_t^y|\mathbf{y}_t, \mathbf{w}^x) \tag{26}$$

By rearranging Eq. (23) and Eq. (24), we can see that DIFFUSEQ can be seen as a more generalized form of iter-NAR before marginalizing out $\{\mathbf{y}_T, \ldots, \mathbf{y}_0\}$, where Eq. (23) and Eq. (24) are equivalent with different computation order, despite the different initialization of $\mathbf{y}_T$. For NAR models, $\mathbf{y}_T$ is uniform copied from the source sentence or *unk*'s token embedding (Gu et al., 2018); for diffusion models, $\mathbf{y}_T$ is sampled from normal distribution $\mathcal{N}(0, \mathbf{I})$.

It is notable that unlike AR and fully NAR models generating text all at once, iterative NAR and diffusion models feature a self-corrected text generation process. The graphical comparison is shown in Figure 7.

## C   FROM DIFFUSEQ TO ITERATIVE NAR AND DIFFUSION MODELS

**From DIFFUSEQ to Iterative NAR**   We show how to derive DIFFUSEQ to iterative non-autoregressive model on discrete space.

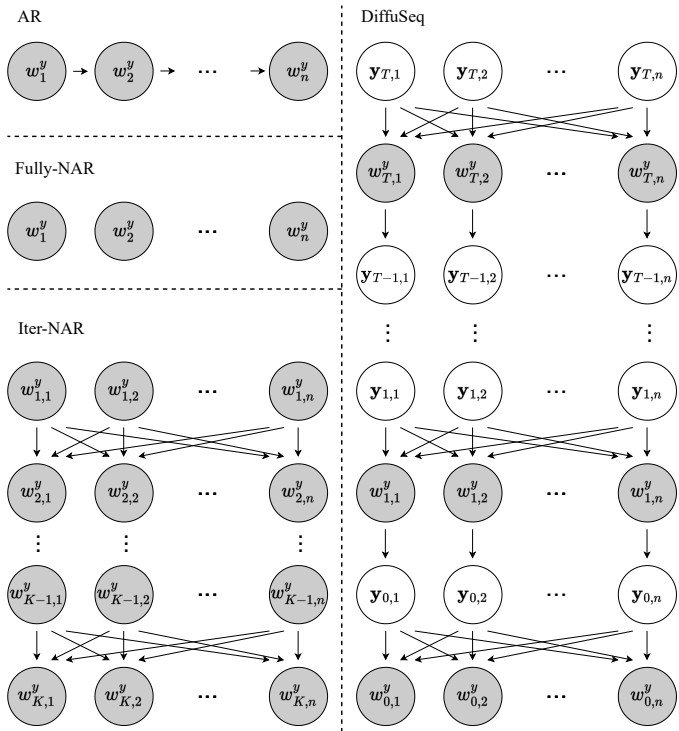

Figure 7: Graphical Models of AR, Fully NAR, iterative NAR and DIFFUSEQ models. For simplicity, we omit source node $\mathbf{w}^x$. Gray nodes indicate dependency on the source node while white nodes are independent to the source node.

$$p_{\text{DIFFUSEQ}}(\mathbf{w}^y|\mathbf{w}^x)$$

$$= \sum_{\mathbf{w}_T^y,\ldots,\mathbf{w}_1^y} \int_{\mathbf{y}_T,\ldots,\mathbf{y}_0} p(\mathbf{w}^y|\mathbf{y}_0,\mathbf{w}^x) \prod_{t=T,\ldots,1} p(\mathbf{y}_{t-1}|\mathbf{w}_t^y)p(\mathbf{w}_t^y|\mathbf{y}_t,\mathbf{w}^x)$$

$$= \sum_{\mathbf{w}_T^y,\ldots,\mathbf{w}_1^y} \int_{\mathbf{y}_T,\ldots,\mathbf{y}_0} p(\mathbf{w}_T^y|\mathbf{y}_T,\mathbf{w}^x) \prod_{t=T-1,\ldots,0} p(\mathbf{w}_t^y|\mathbf{y}_t,\mathbf{w}^x)p(\mathbf{y}_t|\mathbf{w}_{t+1}^y) \quad \text{reorder computation}$$

$$= \sum_{\mathbf{w}_T^y,\ldots,\mathbf{w}_1^y} p(\mathbf{w}_T^y|\mathbf{y}_T,\mathbf{w}^x) \prod_{t=T-1,\ldots,0} \int_{\mathbf{y}_t} p(\mathbf{w}_t^y|\mathbf{y}_t,\mathbf{w}^x)p(\mathbf{y}_t|\mathbf{w}_{t+1}^y)$$

$$= \sum_{\mathbf{w}_T^y,\ldots,\mathbf{w}_1^y} p(\mathbf{w}_T^y|\mathbf{y}_T,\mathbf{w}^x) \prod_{t=T-1,\ldots,0} p(\mathbf{w}_t^y|\mathbf{w}_{t+1}^y,\mathbf{w}^x)) \quad \text{marginalize over } \mathbf{y}$$

$$= \sum_{\mathbf{w}_1^y,\ldots,\mathbf{w}_{K-1}^y} p(\mathbf{w}_1^y|\mathbf{w}^x) \prod_{k=1\ldots K-1} p(\mathbf{w}_{k+1}^y|\mathbf{w}_k^y,\mathbf{w}^x) \quad \text{align } t \text{ and } k \text{ reversely.}$$

$$= p_{\text{iter-NAR}}(\mathbf{w}^y|\mathbf{w}^x)$$

**From DIFFUSEQ to diffusion model** We show how to derive DIFFUSEQ to the straight-forward diffusion model on continuous space.

$$p_{\text{DIFFUSEQ}}(\mathbf{w}^y|\mathbf{w}^x)$$

$$= \sum_{\mathbf{w}_T^y,\ldots,\mathbf{w}_1^y} \int_{\mathbf{y}_T,\ldots,\mathbf{y}_0} p(\mathbf{w}^y|\mathbf{y}_0,\mathbf{w}^x) \prod_{t=T,\ldots,1} p(\mathbf{y}_{t-1}|\mathbf{w}_t^y)p(\mathbf{w}_t^y|\mathbf{y}_t,\mathbf{w}^x)$$

$$= \int_{\mathbf{y}_T,\ldots,\mathbf{y}_0} p(\mathbf{w}^y|\mathbf{y}_0,\mathbf{w}^x) \prod_{t=T,\ldots,1} \sum_{\mathbf{w}_t^y} p(\mathbf{y}_{t-1}|\mathbf{w}_t^y)p(\mathbf{w}_t^y|\mathbf{y}_t,\mathbf{w}^x)$$

$$= \int_{\mathbf{y}_T,\ldots,\mathbf{y}_0} p(\mathbf{w}^y|\mathbf{y}_0,\mathbf{w}^x) \prod_{t=T,\ldots,1} p(\mathbf{y}_{t-1}|\mathbf{y}_t,\mathbf{w}^x) \qquad \textit{marginalize over } \mathbf{w}^y$$

$$= p_{\text{diffusion}}(\mathbf{w}^y|\mathbf{w}^x)$$

## D  DETAILS OF EXPERIMENTS

### D.1  PROCESSING OF QUESTION GENERATION DATASET

To construct high-quality document-question pairs from the Quasar-T dataset, which consists of $\langle document_i, question, answer \rangle$ triplets, we extract $\langle document_i, question \rangle$ pairs if $answer$ exactly matches $document_i$. After pre-processing, we obtain 119K document-question training pairs.

### D.2  SETTINGS OF BASELINES

We compare the settings of different models, including the number of parameters and how to sample the different output sentences, as shown in Table 4. For plain GRU-based encoder-decoder methods, we do not implement diversity search algorithms on it, thus its sentence-level diversity could be very poor. For NAR-LevT, we set max iteration to 9 and follow the termination condition mentioned in the original paper. For GPVAE-T5, we tune the scalar to find the best trade-off between quality and diversity on the dev set. The scalars of all four tasks are set to 2. We implement GPT2 baselines using HuggingFace `Transformers` and for the baseline Transformer-base, we use `Fairseq`.

Table 4: The comparison for different models

| Models | # Parameters | Learning Paradigm | Diversity Source |
|---|---|---|---|
| GRU-attention | 65M | encoder-decoder | - |
| Transformer-base | 80M | encoder-decoder | Temperature/DBS |
| GPT2-base FT | 117M | pretrain-finetune | Hybrid strategy[4] |
| GPT2-large FT | 774M | pretrain-finetune | Hybrid strategy |
| GPVAE-T5 | 220M | pretrain+VAE | Gaussian sampling |
| NAR-LevT | 80M | non-autoregressive | - |
| DIFFUSEQ | 91M | non-autoregressive | Gaussian sampling |

### D.3  DIVERSITY AND QUALITY TRADE-OFF SETTINGS

We list the details to obtain Figure 4. For GPVAE-T5, we set different scalars as $1, 2, 3, 4$. For DIFFUSEQ, we choose trained models at different training steps to achieve different trade-off points. For other baselines, there is no explicit factor to control the diversity generation, so we leave them as single points in the figure.

---

[4]Including top-p sampling, temperature, diversity beam search (DBS) and etc. Implement using HuggingFace `Transformers` https://github.com/huggingface/transformers

## D.4 METRICS

The used BLEU score is sentence-level smoothed from BLEU-1 to 4, and used ROUGE-L score is longest common subsequence based statistics. The implementation is based on `NLTK`[5] and `torchmetrics`. The n-gram based metrics may fail to capture the semantic meaning of sentences, so we consider using BERTScore[6]. Specifically, we use `microsoft/deberta-xlarge-mnli` to help BERTScore correlate better with human scores.

## D.5 GENERATION RESULTS

For different tasks we list some generation examples. As we can see in Table 5, Table 6 and Table 7, DIFFUSEQ tends to generate diverse outputs, but sometimes the sentence is not as fluent as finetuned GPT2.

Table 5: Sample outputs with different random seed in Dialogue test set.

***Utterance***: *How long does the dye last?*
***Response***: *Just did this two days ago, not sure how it'll fade yet!*

| GPVAE-T5 | NAR-LevT |
|---|---|
| * I'm not sure, I'm not sure. I've tested it a few times, but I don't know for sure. I've | * half . |
| * I'm not sure. I'm not sure how long it lasts, I'm sure it 'll get better. It's been a while since | * half . |
| * I've been using it for about a year and a half. I've been using it for about a year and a half. | * half . |
| **GPT2-large finetune** | **DIFFUSEQ** |
| * Two weeks in my case. | * About an hour, 5 days or so. |
| * I've had it for about a year. | * 4 days. |
| * The dye can sit around for a month then you can wash it. | * I'm not sure about this, about the same kind of time. |

Table 6: Sample outputs with different random seed in Question Generation test set.

***Statement***: *The Japanese yen is the official and only currency recognized in Japan.*
***Question***: *What is the Japanese currency?*

| GPVAE-T5 | NAR-LevT |
|---|---|
| * What is the japanese currency | * What is the basic unit of currency for Japan ? |
| * What is the japanese currency | * What is the basic unit of currency for Japan ? |
| * What is the japanese currency | * What is the basic unit of currency for Japan ? |
| **GPT2-large finetune** | **DIFFUSEQ** |
| * What is the basic unit of currency for Japan? | * What is the Japanese currency |
| * What is the Japanese currency | * Which country uses the "yen yen" in currency |
| * What is the basic unit of currency for Japan? | * What is the basic unit of currency? |

---

[5] https://www.nltk.org/_modules/nltk/translate/bleu_score.html
[6] https://github.com/Tiiiger/bert_score

Table 7: Sample outputs with different random seed in Text Simplification test set.

**Complex sentence**: *People can experience loneliness for many reasons, and many life events may cause it, such as a lack of friendship relations during childhood and adolescence, or the physical absence of meaningful people around a person.*
**Simplified**: *One cause of loneliness is a lack of friends during childhood and teenage years.*

| **GPVAE-T5** | **NAR-LevT** |
|---|---|
| * People can experience loneliness for many reasons, and many life events may cause it, such as a lack of friendship relations during childhood and adolescence, or the physical absence of meaningful people around a person | * People may experience reashapphapphapphapphapphappabout life reasit. |
| * People can experience loneliness for many reasons, and many life events may cause it, such as a lack of friendship relations during childhood and adolescence, or the physical absence of meaningful people around a person | * People may experience reashapphapphapphapphapphappabout life reasit. |
| * People can experience loneliness for many reasons, and many life events may cause it, such as a lack of friendship relations during childhood and adolescence, or the physical absence of meaningful people around a person | * People may experience reashapphapphapphapphapphappabout life reasit. |
| **GPT2-large finetune** | **DIFFUSEQ** |
| * Loneliness can be caused by many things. | * Many life events may cause of loneliness |
| * Loneliness can affect people in many ways. | * People can also be very experience loneliness for many reasons. |
| * Loneliness can be caused by many things. | * People can experience loneliness for many reasons, and many life events may, cause it. |

