# OpenReview forum: "DiffuSeq: Sequence to Sequence Text Generation with Diffusion Models"
_ICLR.cc/2023/Conference — ICLR 2023 poster_

### Official Review · Reviewer_juT2 · 2022-10-18

**Confidence:** 4
**Correctness:** 2
**Technical Novelty And Significance:** 2
**Empirical Novelty And Significance:** 3
**Recommendation:** 5

**Clarity, Quality, Novelty And Reproducibility:**

The paper is generally clear and easy to follow, except for a few unconventional word choices (see example below).
As for the reproducibility, it is difficult to assess without the supplementary code; assuming it contains the proper instructions and dependencies, it should be fairly easy to make all experiments reproducible. The two exceptions are (1) missing hardware specifications for inference speed and (2) the type of BLEU used for evaluation (see below).

> This modification (termed partially noising) allows us to cope diffusion models with conditional language modeling.

To the best of my understanding, this is an unconventional use of "cope". One alternative would be consider "adapt __ for __". This is a minor concern that does not affect my score.


> we use the standard metric BLEU (Papineni et al., 2002)

For the sake of reproducibility, please specify the exact version of BLEU you used. For instance, SacreBLEU [4] can be different from NLTK's BLEU [5], which is in turn different from the implementation in MOSES[6]. This is a minor concern that does not affect my overall recommendation.


### Questions


> Diversity Source : Temperature (Table 4)

- did you use sampling with temperature when reporting the quality of autoregressive models? If so, how does it change when using beam search? (if needed, see FairSeq or Hugging Face Transformers for fairly standard beam search implementations).
- how does DiffuseSeq compare against diversity methods based on diverse beam search[9]?


> We first generate a set of candidate samples S from different random seeds of DIFFUSEQ and select the best output sequence that achieves the minimum expected risk under a meaningful loss function (e.g. BLEU or other cheaper metrics like precision). In practice, we use the negative BLEU score in our implementation

Please clarify: does this reranking strategy require knowing target sequence? If not, please explain how exactly is BLEU computed (assuming this is a sentence-level BLEU, what is the target?). Do you apply this strategy for AR / NAR baselines as well - and if so, how do you do that?



[4] https://github.com/mjpost/sacrebleu

[5] https://www.nltk.org/

[6] http://www2.statmt.org/moses/

[7] https://github.com/facebookresearch/fairseq

[8] https://github.com/huggingface/transformers

[9] https://arxiv.org/abs/1610.02424

**Strength And Weaknesses:**


### Strengths

- the proposed method is evaluated in multiple seq2seq tasks, supporting the claim for general seq2seq modeling
- authors attempt to show the connection between DiffuSeq and existing seq2seq strategies
- the paper is reasonably well-written and easy to follow

### Weaknesses

I have a single main concern, justifying my current (negative) recommendation -- the theoretical need for a new type of denoising diffusion specific to Seq2Seq problems. The other weaknesses i found are fairly minor and are listed at the bottom.

As authors note in the paper, prior works have already adapted denoising diffusion to unconditional language modeling. To extend their results to seq2seq problems, one must find a way to incorporate additional inputs to the diffusion model. In continuous "text-to-image" diffusion, this can be done by simply feeding the extra inputs to the model.
Modern diffusion models for conditional image generation, such as Imagen[1] or DALLE-2 [2] simply attend to the input text inside the diffusion model __without any changes to the underlying diffusion or noise distribution__.

Hence, my main question: __does DiffuSeq really need the extra complexity of "partially noising", instead of just treating x as inputs?__ To the best of my understanding, the x part is not generated by the model, but frozen throughout the diffusion process. As a result, the model does not learn to denoise it, and can effectively treat it as an extra input to the existing (unconditional) text diffusion methods. Please correct me if my understanding is flawed in any way.

If DiffuSeq does *not* need the extra math, then this paper effectively becomes an empirical study of existing diffusion models for seq2seq applications, which leads to my __second concern: non-standard evaluation tasks__. As authors state in the paper, seq2seq is a very popular and competitive area of research with many popular datasets. However, the datasets/tasks chosen in the paper are not among the popular ones (see [3]), which makes it difficult to assess whether the baseline models are state-of-the-art for those tasks.



[1] https://arxiv.org/abs/2205.11487

[2] https://cdn.openai.com/papers/dall-e-2.pdf

[3] https://paperswithcode.com/area/natural-language-processing

**Summary Of The Paper:**

The paper studies the application of denoising diffusion probabilistic models to seq2seq text generation. Authors propose DiffuSeq - a diffusion-based model for seq2seq task using a technique they call "partially noising". Authors also show the theoretical connection between DiffuSeq and standard AR / NAR  language models. They evaluate DiffuSeq on a number of seq2seq tasks such as dialogue generation and paraphrasing, against several autoregressive and non-autoregressive baselines, achieving comparable generation quality. On top of that, authors note that DiffuSeq generates more diverse outputs. Authors analyze the behavior of the proposed model, report inference throughput and ablate several design decisions.

**Summary Of The Review:**

I left a generally negative review, mainly because I do not see the need for the extra complexity of DiffuSeq, compared to simply treating the source text x as an extra input. My concern is based on how input text is treated in existing diffusion-based text-to-image models[1,2].
Outside of that, the paper appears to be generally decent, with both strong and weak points as described above. I am open to discussion and, _should authors prove that my main concern is wrong_, increasing my score.

---

> ### Author Response · Authors · 2022-11-10
> **Reponse to Reviewer juT2 (I)**
>
> Thank you for your valuable comments and suggestions that will help make our article clearer. We summarize your main concerns as follows:
>
> >Q1: Does DiffuSeq really need the extra complexity of "partially noising", instead of just treating x as inputs?
>
> A1: We guess there may be some misunderstandings about "partially noising". Let's futher explain it here. In our implementation, we actually take $\mathbf{x}$ as inputs along with $\mathbf{y}$ (given $\mathbf{z}=\mathbf{x} \oplus \mathbf{y}$), the noise is added only on the $\mathbf{y}$. The information of $\mathbf{x}$ kicks in through the self-attention mechanism. **We didn't add extra complexity to the underlying diffusion**. However, **it would be over-simplified if we merely ignore the modality difference between image data and text data**. We elaborate the differences between DiffuSeq and the text-to-image diffusion model below:
>
> 1. DiffuSeq needs to reconstruct the discrete text data, which remains a challenging problem[1][2][3], while the text-to-image diffusion model is to restore the continuous images directly. We have the extra rounding operation for discrete text data and hence theoretical modeling is necessary and essential due to the difference between image and text modeling.
> 2. DiffuSeq shares the embedding function between source and target sequences, **enabling the training of two different feature spaces jointly** (and from scratch), while the text-to-image diffusion model is two-staged and relies on a frozen text encoder. We also attempt to fix the input x's embedding vector using pre-trained LM (as you mentioned *treating x as inputs*) and our ablation Table 3 showcases that the joint training for sequence pairs' embedding space is better than fixed x. This sets DiffuSeq away from existing solutions.
>
>   [1] [Structured denoising diffusion models in discrete state-spaces.](https://arxiv.org/abs/2107.03006)
>
>   [2] [Diffusion-lm improves controllable text generation.](https://arxiv.org/abs/2205.14217)
>
>   [3] [Analog bits: Generating discrete data using diffusion models with self-conditioning.](https://arxiv.org/abs/2208.04202)
>
> >Q2: Concern about non-standard evaluation tasks
>
> A2: We do adopt standard evaluation on sequence-to-sequence tasks. The evaluation tasks considered here, including paraphrase[1], text style transfer (text simplification) [2], dialogue generation [3], question generation [4], are all widely used in the field [5]-[8]; and for these tasks, **pre-trained language models (GPT2, T5) are often seen as strong baselines** [9]. QQP is a widely used dataset and for the rest of the three tasks, we choose recently proposed datasets [10][11][12] in favor of larger sizes of these datasets.
>
>   [1] https://paperswithcode.com/task/paraphrase-generation
>
>   [2] https://paperswithcode.com/task/text-style-transfoer
>
>   [3] https://paperswithcode.com/task/dialogue-generation
>
>   [4] https://paperswithcode.com/task/question-answer-generation
>
>   [5] [Paraphrase Generation: A Survey of the State of the Art](https://aclanthology.org/2021.emnlp-main.414.pdf)
>
>   [6] https://github.com/csnlp/Dialogue-Generation#open-domain-dialogue-generation
>
>   [7] [Deep Learning for Text Style Transfer: A Survey](https://aclanthology.org/2022.cl-1.6/)
>
>   [8] [Question Generation for Question Answering](https://aclanthology.org/D17-1090/)
>
>   [9] [Language Models are Unsupervised Multitask Learners](https://d4mucfpksywv.cloudfront.net/better-language-models/language_models_are_unsupervised_multitask_learners.pdf)
>
>   [10] [Denoising Distantly Supervised Open-Domain Question Answering](https://aclanthology.org/P18-1161/)
>
>   [11] [Commonsense Knowledge Aware Conversation Generation with Graph Attention](https://www.ijcai.org/proceedings/2018/0643.pdf)
>
>   [12] [Neural CRF Model for Sentence Alignment in Text Simplification](https://aclanthology.org/2020.acl-main.709.pdf)
>
> >Q3: about reproducibility
>
> A3: We will release the code in two weeks. The hardware specifications for inference speed: on single NVIDIA A100 Tensor Core GPU, already updated it in the revised draft (highlighted in red). We use BLUEScore from torchmetrics, which is a NLTK BLEU implementation (listed in Appendix D.4).
>
> *(Due to page limitation, please move to the next section.)*

---

> > ### Author Response · Authors · 2022-11-10
> > **Reponse to Reviewer juT2 (II)**
> >
> > >Q4: Did you use sampling with temperature when reporting the quality of autoregressive models? If so, how does it change when using beam search?  how does DiffuseSeq compare against diversity methods based on diverse beam search?
> >
> > A4: As mentioned in appendix D.2, we get diverse generation results by sampling with temperature and using top_p/top_k sampling too (named hybrid sampling strategy). We conduct beam search/diverse beam search on AR models, including transformer-base model and GPT2-large finetune. The results are shown in the following table:
> >
> > **Open Domain Dialogue**
> > |Models|Diversity Source|BLUE|ROUGE-L|BertScore|selfBLEU/div-4
> > |:---:|:---|:---:|:---:|:---:|:---:|
> > |Transformer-base|Temperature|**0.0189**|0.1039|0.4781|0.3698/0.6472|
> > |Transformer-base|Beam search|0.0180|0.1002|0.4723|0.6597/0.4737|
> > |Transformer-base|Diverse beam search|0.0180|0.0955|0.4897|0.5016/0.6005|
> > |GPT2-large FT|Hybrid strategy|0.0125|0.1002|**0.5293**|0.0213/0.9938|
> > |GPT2-large FT|Beam search|0.0069|0.1225|0.5029|0.1359/0.9020|
> > |GPT2-large FT|Diverse beam search|0.0087|**0.1275**|0.5144|0.1280/0.8880|
> > |DiffuSeq|Gaussian sampling|0.0139|0.1056|0.5131|**0.0144/0.9971**|
> >
> > **Question Generation**
> > |Models|Diversity Source|BLUE|ROUGE-L|BertScore|selfBLEU/div-4
> > |:---:|:---|:---:|:---:|:---:|:---:|
> > |Transformer-base|Temperature|0.0364|0.1994|0.5334|0.8767/0.4055|
> > |Transformer-base|Beam search|0.0375|0.1966|0.534|0.8055/0.4675|
> > |Transformer-base|Diverse beam search|0.0352|0.1923|0.5161|0.6723/0.4996|
> > |GPT2-large FT|Hybrid strategy|0.1110|0.3215|**0.6346**|0.2910/0.8062|
> > |GPT2-large FT|Beam search|0.0378|0.1999|0.5823|0.8568/0.4141|
> > |GPT2-large FT|Diverse beam search|0.0352|0.1940|0.5788|0.5374/0.6228|
> > |DiffuSeq|Gaussian sampling|**0.1731**|**0.3665**|0.6123|**0.2789/0.8103**|
> >
> > **Text Simplification**
> > |Models|Diversity Source|BLUE|ROUGE-L|BertScore|selfBLEU/div-4
> > |:---:|:---|:---:|:---:|:---:|:---:|
> > |Transformer-base|Temperature|0.2445|0.5058|0.7590|0.8632/0.4028|
> > |Transformer-base|Beam search|0.2359|0.5023|0.7505|0.6941/0.5327|
> > |Transformer-base|Diverse beam search|0.2693|0.4907|0.7381|0.6924/0.5095|
> > |GPT2-large FT|Hybrid strategy|0.2693|0.5111|0.7882|0.6042/0.5876|
> > |GPT2-large FT|Beam search|0.2856|0.5304|0.7970|0.9261/0.3753|
> > |GPT2-large FT|Diverse beam search|0.2474|0.4948|0.7806|0.5493/0.6503|
> > |DiffuSeq|Gaussian sampling|**0.3622**|**0.5849**|**0.8126**|**0.4642/0.6604**|
> >
> > **Paraphrase**
> > |Models|Diversity Source|BLUE|ROUGE-L|BertScore|selfBLEU/div-4
> > |:---:|:---|:---:|:---:|:---:|:---:|
> > |Transformer-base|Temperature|0.0580|0.2489|0.5392|0.7717/0.4312|
> > |Transformer-base|Beam search|0.0549|0.2468|0.5455|0.4705/0.7239|
> > |Transformer-base|Diverse beam search|0.0557|0.2477|0.5350|0.6004/0.5731|
> > |GPT2-large FT|Hybrid strategy|0.2059|0.5415|0.8363|0.7325/0.5020|
> > |GPT2-large FT|Beam search|0.2119|0.5480|**0.8389**|0.9673/0.3495|
> > |GPT2-large FT|Diverse beam search|0.1679|0.4821|0.8013|0.3083/0.7812|
> > |DiffuSeq|Gaussian sampling|**0.2413**|**0.5880**|0.8365|**0.2732/0.8641**|
> >
> > Experimental results show that for GPT2-large finetune model, beam search can rarely improve the diversity compared with the sampling strategy. We assume that it is because the tokens' log probability is more long-tailed for PLMs, so it is harder for beam search to search more diverse results than the sampling strategy which directly softens the log distribution. Diverse beam search helps to promote the diversity score but sacrifices the quality score. Similarly, for the transformer-base model, although beam search/diverse beam search can get more diverse results, it still suffers from quality-diversity trade-off. **In the end, DiffuSeq still achieves better diversity results and remains competitive generation quality**.
> >
> > >Q5: Please clarify: does this reranking strategy require knowing target sequence?
> >
> > A5: Of course, our reranking strategy **DO NOT require knowing target sequence**. It is based on Minimum Bayes Risk (MBR) using BLEU score. For example, if we have three sentences in the candidate set $\\{a, b, c\\}$, the final score of sentence $a$ is: $1/2(BLEU(a, b)+BLEU(a, c))$, and it is the same for $b$ and $c$. In the end, we will choose the sentence with the highest score.
> >
> > >Q6: Do you apply this strategy for AR / NAR baselines as well - and if so, how do you do that?
> >
> > A6: In the main Table 1, we didn't apply this strategy to AR/NAR models, but **we apply it to GPT2 in Fig 3**, using **the same MBR strategy** (refer to Q5). According to Fig 3, we can speculate that DiffuSeq benefits from the reranking mainly due to its highly diversified generated sentences, and other methods like AR/NAR with less diversified candidate sets may not benefit much from MBR.

---

> ### Author Response · Authors · 2022-11-28
> **Looking forward to receiving your feedback**
>
> Dear Reviewer juT2,
>
> In the previous section, we respond to your two main concerns and add additional quantitative results. We would be very grateful if you could provide feedback on our rebuttal. If there are any additional questions, please feel free to ask us. We will immediately respond to them.
>
> We understand you are very busy and appreciate your time. We will be waiting for your valuable feedback.
>
>
> Best wishes,
>
> Authors

---

> > ### Comment · Reviewer_juT2 · 2022-12-06
> > **Response to author rebuttal**
> >
> > I acknowledge the authors comments. With additional evaluations and clarifications in mind, i increase my overall recommendation to 5.
> >
> > The two concerns left unresolved are:
> >
> > 1. On partially noising: my original concern
> >
> > On response part 1: I recognize that generating discrete data is different from continuous one - and that was not my concern. However, (a) prior work already proposes means to use diffusion for generating discrete data unconditionally and (b) I argue that merely adding a condition
> >
> > On response part 2: To the best of my understanding, text2image diffusion uses frozen text encoder to improve performance, but this is not mathematically required by the model. Similarly, you could have started with a pre-trained BERT or similar model as an encoder without significant changes to the optimization problem.
> >
> > 2. On evaluation tasks.
> >
> > While I admit that these tasks fit a certain definition of "standard". Still, they are far from the most popular evaluation tasks, and, to the best of my knowledge, there is no justification for taking them over the popular ones. While the evaluations are still scientifically valid and support the claims, it would be significantly easier for readers to quantify the advantage of diffuseq on more popular seq2seq tasks (such as machine translation, or even summarization; see my original concern). To clarify: I do not claim that these tasks are inherently superior or more important, but there are objectively more works and more established evaluation benchmarks.

---

> > > ### Author Response · Authors · 2022-12-08
> > > **Thanks again for your review**
> > >
> > > More discussions:
> > >
> > > 1.1 We believe that integrating the source sentence as a condition in a classifier-free manner is an effective method for dealing with seq2seq tasks, particularly those that require diverse generations. To the best of our knowledge, **our study is the first successful attempt to apply diffusion model to seq2seq task**, and we present several useful strategies, such as joint embedding training and importance sampling, to facilitate its use in these tasks.
> > >
> > > 1.2  In Table 3, we use a frozen text encoder (the 128d embedding of pretrained tiny-BERT) to encode the input x, with the same setting as text2img diffusion models. However, the performance of this setting is inferior to that of joint training in DiffuSeq. While it may be possible to utilize a more advanced frozen text encoder, we believe that the challenge of recovering image data versus text data from latent vector space is not equivalent. **Our experimental results suggest that the use of a fixed text encoder may not be the optimal solution for seq2seq tasks**, reinforcing our argument that our proposed method is a superior solution for these tasks.
> > >
> > > 2 While it is true that machine translation and summarization are more widely studied seq2seq tasks, **DiffuSeq was developed to leverage the benefits of diffusion models, specifically the diversity of generation.** This is our starting point. In fact, the diversity of generation is not a primary concern in MT and summarization, as these tasks prioritize accuracy over variety. Therefore, we chose to focus on open-text generation tasks such as open-domain dialogue generation and paraphrasing, where diversity can provide greater benefits.
> > >
> > > Thank you for your reply and for sharing your insights.
> > >
> > > Best,
> > >
> > > Authors

---

### Official Review · Reviewer_jTcw · 2022-10-23

**Confidence:** 2
**Correctness:** 4
**Technical Novelty And Significance:** 3
**Empirical Novelty And Significance:** 3
**Recommendation:** 6

**Clarity, Quality, Novelty And Reproducibility:**

The paper has an overall clear structure. The problem solved by the papers is important; The method is sufficiently novel and the results can be clearly reproduced. It would be better to state computational requirements in the main text.
Writing:
- "Partially noising" -> "partial noising"
- Notation at the beginning of Sec.3 is not self-contained and needs refinement. Particularly, z was never introduced. The admitted abuse of notation in the absence of properly introduced notation does not look good.
- Fig. 2 was never referred to for method illustration purposes (the first sentence of Sec. 3 doesn't count as it doesn't list any details)
- "cope" misused around the same paragraph
- "standard variation" sounds like an exotic term, I'm not sure if it is any better than what already exists, standard deviation, variance. For the case of multivariate Gaussian distribution, diagonal covariance would also be an appropriate term.


**Strength And Weaknesses:**

The paper leaves good overall impression due to high-quality writing, propose positioning against state of the art, and good empirical study. Story-telling could be improved with proper background/notation introduction, clearing any abuse of notation, offloading any derivations to supplementary, and focusing on key findings and theoretical conclusions being drawn. Particularly, "connections" paragraph seems to be overly detailed, yet Eq. 1 and 2 are neither too detailed nor distilled to the barebones. I appreciate the author's usage of equation annotations, yet I found them hard to follow anyway.

**Summary Of The Paper:**

The paper introduces conditional diffusion models in the NLP setting, demonstrates its empirical advantages, and draws theoretical connections with autoregressive and iterative-nonautoregressive models.

**Summary Of The Review:**

Overall a good paper with few loose ends, but writing, clarity, and story-telling could be improved.

---

> ### Author Response · Authors · 2022-11-10
> **Response to Reviewer jTcw**
>
> Thank you for your suggestions and we have updated some writings in the revised draft (highlighted in red). We believe that the analysis of the language modeling capacity among different generative models in "connections" paragraph is interesting and some experiment results (e.g. Fig 3 and Fig 5) also correspond to this section. But considering the page limitation, we will rearrange page 4 and put details in Appendix.

---

### Official Review · Reviewer_LMuu · 2022-10-25

**Confidence:** 4
**Correctness:** 3
**Technical Novelty And Significance:** 3
**Empirical Novelty And Significance:** 3
**Recommendation:** 8

**Clarity, Quality, Novelty And Reproducibility:**

Most parts of the paper are clear and the novelty is good. No code provided and the reproducing ability is average.

**Strength And Weaknesses:**

Strong:

1. Diffuseq framework, with new objective for training and concatenates x (given) and y (target) for diffusion model training and consequent inferencing;

2. Better results compared with a list of strong baselines.

Weak:

1. Different with the paper’s arguments, it is a bit hard to find the significant difference in terms of technical novelty with existing diffusion-LM – concatenating x and y actually almost doubles the diffusion space and possibly makes the training/inferencing speed to be much slower than diffusion-lm – a detailed experimental comparison with diffusion-lm is required.

2. Some essential parts need to be enriched – such as equation 2. And most equations in page 4 are less important to be included – give the spaces to equation 2 – which is your novel objective function. (appendix A’s extension is far from enough, such as how EMB’s item square was removed).


Detailed questions and comments

1. Figure 1, the t=T/2 figures are real examples? Frequently in my experiments, the figures are clear only at the last several steps and did not achieve the clear results at t=T/2 steps.
2. Also in figure 1, can you make it clearer the difference of “guidance” in diffuseq vs. “condition” in diffusion-LM – align these differences in figures with the specific equations’ differences.
3. Equation 2, more details are preferred and even appendix A is not clear enough of from the two equations: such as how the EMB included item changed from square to without square. I could not find the details in appendix A.
4. Page 4’s equations are less important (most are not novel points of this paper) and I think they can be reduced or part of them can be moved to appendix – leaving more spaces for explaining equation 2 – which is your own objectives.
5. The “1” involved in equation 2 is a bit ambiguous. Basing on current description, I think it is t=1, not a simple 1 and prefer to see how exactly you implemented this in your code – did you use t=[0, 1] or t=[0, T]?
6. Table 1, can you also give the results of diffusion-LM? More than half baselines selected here are less comparable than existing diffusion-based generation models.
7. In “inference speed”, “1000 diffusion steps” a similar inference speed. I am not able to understand this part. Any detailed reasons on this? How about the speed comparison on other tasks besides “question generation”? does the same speech comparison results hold?
8. Table 3, right-bottom item, “0.9376” is larger than “0.8103” and why this happen? Any detailed error analysis on this? If you say diffusion-seq is strong at diversity, then this result actually did not align with that.





**Summary Of The Paper:**

This paper proposes diffuseq – a diffusion model for sequence-to-sequence text generation tasks. The x and y pairs are concatenated together and sent to the forward diffusion process. Different with diffusion-lm’s classifier-guided diffusion, diffuseq here uses classifier-free diffusion guided by points in space.  Similar with diffusion-lm, an embedding on words is earned jointly with the diffusion/reverse processes.
Experiments on rich text generation tasks show that the proposed diffuseq could achieve better results than strong pretrained language model baselines or vae variant baselines.


**Summary Of The Review:**

The concatenate of x and y together for diffusion is the major novel part with a well-designed objective function. Most parts of the paper are well written. The score will for sure be improved if equation 2 can be enriched and more comparison with strong baselines such as diffusion-lm can be given here.

This paper is above the average level and higher scores are deserved with the updating of the current unclear parts.

--

after reading the authors' feedbacks, I decide to rank higher this paper. Basically, the general idea of seq2seq with diffusion is an interesting direction and many related seq2seq tasks should benefit from it. It is good to see that the authors enriched their core ideas and replace some less important parts of page 4 to the appendix, hopefully, this makes this paper easier to be followed.

[still, a bit doubt about that diffusion-LM can not deal with seq2seq problems since in Diffusion-LM's paper, a number of controllable text-to-text generation tasks were reported... anyway, this only influence the comparison part of this paper, not the central idea of this seq2seq-diffusion models.]

---

> ### Author Response · Authors · 2022-11-10
> **Response to Reviewer LMuu**
>
> Thank you for your valuable comments and suggestions that will help make our article clearer. We summarize your main concerns as follows:
>
> >Q1: It is a bit hard to find the significant difference in terms of technical novelty. Concatenating x and y actually almost doubles the diffusion space and possibly makes the training/inferencing speed much slower than Diffusion-LM.
>
> A1: Diffusion-LM targets different generation scenarios from DiffuSeq. Diffusion-LM adds constraints on the output sentence of models, **which can not handle Seq2Seq tasks**, while DiffuSeq generates sentence y based on input sentence x, so we cannot directly compare them for performance and speed. Technically, Diffusion-LM is two-staged and relies on **extra-trained-classifiers** to compose fine-grained constraints where extra time is needed too, while DiffuSeq is end-to-end and **classifier-free**.
>
> >Q2: Some essential parts need to be enriched (Eq.2), and page 4’s equations are less important. How the EMB included item changed from square to without square in Eq.2.
>
> A2: We will rearrange page 4 and put details in Appendix. We will also enrich Eq.2. In Eq.2 there was a typo, where EMB’s item square should not be removed, and we have updated it in the revised draft (highlighted in red). Hope the revised version could dispel your concerns. It should be:
>  $\min\_{\theta}\left[ \sum\_{t=2}^T||\mathbf{y}\_0-\tilde f\_{\theta}(\mathbf{z}\_t, t)||^2 + ||Emb(\mathbf{w}^y)-\tilde f\_{\theta}(\mathbf{z}\_1, 1)||^2 + \mathcal{R}(||\mathbf{z}\_0||^2)\right]$,
> where $||\mathbf{y}\_0-\tilde f\_{\theta}(\mathbf{z}\_t, t)||^2$ corresponds to $\mathcal{L}\_t$ loss while the $||Emb(\mathbf{w}^y)-\tilde f\_{\theta}(\mathbf{z}\_1, 1)||^2$ corresponds to $\mathcal{L}\_0$. The regularization term $\mathcal{R}(||\mathbf{z}\_0||^2)$) helps to regularize the embedding learning.
>
> >Q3: In Figure 1, the t=T/2 figures are real examples? Can you make it clearer the difference of “guidance” in DiffuSeq vs. “condition” in Diffusion-LM?
>
> A3: Fig1 is not using real examples, and it is just an illustration of the differences between different kinds of models. In experiments, we have the same findings as yours, the text is **clear only at the last several steps** and did not achieve clear results at t=T/2 steps (**Please refer to the orange dot line in Figure 5**).
>
> A clearer description of the difference: the guidance in DiffuSeq is built-in, entangled with the diffusion model and trained in **classifier-free** manner, while the Diffusion-LM is to first train an unconditional language model and use **extra-classifiers** to constraint the outputs.
>
> >Q4: "1" in Eq.2 is a bit ambiguous.
>
> A4: "1" is indeed the step=1 instead of a simple 1. We use t=[0, 1] in implementation. We will release the code soon.
>
> >Q5: Can you also give the results of Diffusion-LM in Table 1? Baselines selected here are less comparable than existing diffusion-based generation models.
>
> A5: Diffusion-LM can not tackle Seq2Seq tasks as DiffuSeq does (please refer to Q1), and among existing diffusion-based generation models, we are the first to deploy it on Seq2Seq text generation. For these tasks, **pre-trained language models (GPT2, T5) are often seen as strong baselines**.
>
> >Q6: Any detailed reasons on the "inference speed" in Fig. 6?
>
> A6: In Fig. 6, and the x-axis should be the "sampling steps" instead of "diffusion steps" in original paper. Here "diffusion steps" means using different training steps while keeping the same steps with training when sampling, and "sampling steps" means that we fix the training steps to 2000 but use DDIM to shrink the sampling steps to 1000, in order to speed up the sampling process. We wrongly put the "diffusion steps" figure but actually we want to show the results of "sampling steps". The correct data should be like this:
>
> |  Sampling steps |  1000   |  2000  |
> |:--------|:---------|:--------|
> | BLEU                 | 0.1410 | 0.1567|
> | Speed (items/s) | 1.47      | 0.55   |
>
> Compared with GPT-2 large speed 2.81 items per second and BLEU 0.1110, we hence draw the conclusion that through DDIM sampling DiffuSeq **narrows** the inference speed with GPT2-large while achieving a higher BLEU score. **The same speed comparison results still hold** because for different datasets we use the same sequence length and they hence share similar inference speed.
>
> >Q7: Table 3 right-bottom item, “0.9376” is larger than “0.8103” and why this happen?
>
> A7: In Table 3,  we compare DiffuSeq's the joint training strategy with the decoupled training strategy, where the latter results in much worse generation quality than the former, although it has a better div-4 score. But in fact, if using the latter strategy, the model doesn't yield meaningful results (with a poor BLEU score), and therefore **the high div-4 score based on poor BLEU means little**. This result can be seen as an extreme case of trade-off between quality and diversity shown in Figure 6.

---

> ### Author Response · Authors · 2022-11-28
> **Looking forward to receiving your feedback**
>
> Dear Reviewer LMuu,
>
> In the previous section, we elucidate the differences between DiffuSeq and Diffusion-LM and update unclear parts. We are wondering if there are any additional questions. Please feel free to discuss with us. If you are satisfied with our response and changes, please consider updating your rating.
>
> Best wishes,
>
> Authors

---

> > ### Comment · Reviewer_LMuu · 2022-12-08
> > **thanks for the responses**
> >
> > Sure, thanks the authors for making it clearer and I just ranked higher.
> >
> > Still, frankly speaking, I doubt if diffusion-LM can not be applied in seq2seq tasks or possibly some seq2seq tasks can not be covered by diffusion-LM. Anyway, it does not influence the general novelty of this paper on seq2seq diffusion.

---

> > > ### Author Response · Authors · 2022-12-08
> > > **Thanks again for your review**
> > >
> > > More discussions about Diffusion-LM:
> > > If we consider simple text revision as a seq2seq task, such as revising the sentiment of a sentence from positive to negative or inserting a certain word, it might be possible to apply Diffusion-LM's classifier as guidance. However, we believe that Diffusion-LM is not capable of handling the tasks we have conducted, particularly complex tasks involving open dialogue generation and question generation.
> > >
> > > We are grateful for your thoughtful response. Thank you for sharing your insights.
> > >
> > > Best,
> > >
> > > Authors

---

### Official Review · Reviewer_NoKK · 2022-10-25

**Confidence:** 4
**Correctness:** 3
**Technical Novelty And Significance:** 2
**Empirical Novelty And Significance:** 4
**Recommendation:** 8

**Clarity, Quality, Novelty And Reproducibility:**

I found the details of this paper difficult to parse. There is a lot of notation, much of which is introduced informally. I had difficulty tracking the meaning and use of various symbols. I think I more or less understand the method that has been implemented in this work, but I gave up on understanding the discussion on Page 4: this discussion is tangential to the methods proposed and evaluated in the rest of the paper, however, I do imagine that this discussion could be interesting and valuable with some improvements to the writing.

Notable examples of notational and writing issues (non-exhaustive):

  - The framing of the first sentence of the introduction is weird. Two issues with GAN training (instability and mode collapse) are attributed collectively to broader generative modelling techniques (GAN + VAE + Flow). And the mention of surrogate objectives here is also a little odd, because diffusion models also rely on surrogates (the variational lower bound). Furthermore, autoregressive models are not mentioned. So overall, this initial framing could really use an overhaul.

  - In Section 2 we are introduced to the latent diffusion states using variable "x". However, in Section 3 "x" is used to denote the conditioning sequence, and "z" is used to denote the latent states. Not a big deal, but redefining notation like this is easily avoidable.

  - Given z_t = x_t + y_t, we are told that "we only impose noising on y_t" and that "In terms of implementation, we execute an anchoring function that replaces the corrupted x_t with the original x_0." This anchoring operation is never defined, and I am pretty confused about what is actually meant. I think this means that we don't noise x, so x_t = x_0 for all values of t: this makes sense to me, but it is weird to describe this as adding noise and then removing it.

  - I do not understand the new subscripts "K" on "w" introduced for Equation 5 (the equality given in the preceding sentence does not constitute a meaningful definition.

  - In Equation 6, shouldn't the sum be an integral? This is marginalization over continuous Gaussian random variables.

  - In Equation 7 and 8, we are introduced to variables "w_t" (with subscripts "t"). I'm not sure what this is supposed to mean, because "w" have been defined as the noiseless, discrete input sequences. And again I think that perhaps these sums ought to be integrals.

**Strength And Weaknesses:**

Methodologically, this work is a minor extension of Diffusion-LM (Li et al., 2022). Nevertheless, this style of text generation model is quite new, so the empirical work evaluating these models for seq2seq modeling tasks is both novel and interesting as it helps to build the community's emerging understanding of these moels. The chosen collection of seq2seq modeling tasks and datasets are interesting, and the choice of baseline methods for comparison are both strong and thorough.

The empirical evaluation compares DIFFUSEQ's performance in each of 4 tasks to (1) encoder-decoder autoregressive models (both GRU and Transformer) (2) fine-tuned LLM's (GPT2 and T5) and (3) the non-autoregressive Levenshtein transformer. DIFFUSEQ's quantitative performance is strong compared to all baselines, and is even competitive with finetuned pre-trained LLM's: this is impressive given that DIFFUSEQ is a smaller model, trained on much smaller datasets.

I also appreciate the architectural ablation of the use of joint versus separate embeddings for sequence pairs (Table 3). One additional ablation that could be interesting to see is a comparison of DIFFUSEQ trained with and without importance sampling. I believe this is the first paper to apply importance sampling to text diffusion models, and so it would be quite interesting to see the effect of this adjustment.

**Summary Of The Paper:**

This paper constructs a conditional (sequence-to-sequence) generative diffusion model for text: DIFFUSEQ. Methodologically, the work is quite similar to Diffusion-LM (Li et al., 2022) with the notable adoption of importance sampling (following Nichol and Dhariwal, 2021) to improve training efficiency. Experimentally, this work evaluates DIFFUSEQ using automatic quantitative metrics against a strong collection of baseline methods in four conditional generation tasks: dialog, question generation, text simplification, and paraphrasing.

**Summary Of The Review:**

This paper is an interesting empirical study of seq2seq text diffusion models, with some (fixable) issues in the writing.

---

> ### Author Response · Authors · 2022-11-10
> **Response to Reviewer NoKK**
>
> Thank you for your valuable comments and suggestions that will help make our article clearer. We summarize your main concerns as follows:
>
> >Q1: The effect of importance sampling
>
> A1: In our preliminary experiments, we found that DiffuSeq trained without importance sampling would result in insufficient training. In detail, we train an unconditional language model and evaluate the fluency of generated free text using a teacher LM (following Diffusion-LM (Li et al., 2022)), the perplexity with or without importance sampling is from 51.5 to 64.1. Detailed evaluation metrics of this ablation setting could be added in the later version.
>
> >Q2: The initial framing of the introduction and the notations. Page 4 is tangential to the methods.
>
> A2: Thank you for the valuable advice. We will carefully refine the writing accordingly. We believe that the analysis of the language modeling capacity between different generative models on page 4 is interesting and some experiment results (e.g. Fig 3 and Fig 5) also correspond to this section. But considering the page limitation, we will rearrange page 4 and put details in Appendix.
>
> >Q3: Anchoring operation is never defined, and it is weird to describe this as adding noise and then removing it.
>
> A3: Thank you for the valuable advice. Anchoring operation needs further elucidation. In the reverse sampling process, it is deployed on the output part of the model, defined as $\mathbf{x}_t\prime = \mathbf{x}_0$ for all values of $t$, where the model predicted $\mathbf{x}_t\prime$ is not strictly equal to real $\mathbf{x}_t$, so the anchoring function is to bridge the gap of real input signal $\mathbf{x}_0$. In this stage, it is not simply adding noise and then removing it. In order to symmetrically describe this operation, as well as consider the code implementation, we describe this operation first in the training stage, but it is true that this operation mathematically equals $\mathbf{x}_t = \mathbf{x}_0$ (adding noise and removing) when training. We will rearrange this part.
>
> >Q4: The meaning of subscripts "$K$" on "$\mathbf{w}$" in Eq.5
>
> A4: $K$ is the iteration of iter-NAR model, and $\mathbf{w}_k$ is the word sequence of middle results.
>
> >Q5: Sums ought to be integrals in Eq.6, 7, 8.
>
> A5: Thanks for pointing this out. The marginalization is over continuous vector $\mathbf{y}$, so it should be integral instead of sum in the equation. The revised Eq.6 is
> $p\_\text{diffusion}(\mathbf{w}^y|\mathbf{w}^x)=\int\_{\mathbf{y}_{T},\ldots,\mathbf{y}\_0}p(\mathbf{w}^y|\mathbf{y}\_{0},\mathbf{w}^x) \prod\_{t=T,\ldots,1}{p(\mathbf{y}\_{t-1}|\mathbf{y}\_t,\mathbf{w}^x)}$, and it is the same with the Eq.7~8. We have updated it in the revised draft (highlighted in red).
>
>
> >Q6: The meaning of $\mathbf{w}_t$.
>
> A6: We apply the rounding operation in each sampling step $t$, which is to round the continuous vectors $\mathbf{y}$ to discrete $\mathbf{w}_t$ (which is to say that at each sampling step $t$, we reconstruct the discrete word sequence $\mathbf{w}_t$). This operation is called clamping in Diffusion-LM, but we establish the theoretical modeling of text tokens for these continuous space based textual diffusion methods.

---

> > ### Comment · Reviewer_NoKK · 2022-11-27
> > **Thanks**
> >
> > Thank you for these clarifications and your commitment to improving the exposition of this paper.

---

### Decision · Program_Chairs · 2023-01-20

**Decision:**

Accept: poster

**Justification For Why Not Higher Score:**

Lingering issues surrounding whether the methodological contribution is as strong as presented.

**Justification For Why Not Lower Score:**

The empirical study demonstrates clear improvement over strong baselines.

**Metareview: Summary, Strengths And Weaknesses:**

This paper constructs a conditional (sequence-to-sequence) generative diffusion model for text: DIFFUSEQ. The paper evaluates the method on four tasks in comparison with six baselines, demonstrating the empirical improvement of Diff-Seq2seq. They also draw theoretical connections with autoregressive and iterative-nonautoregressive models.

**Strengths:**

The reviewers generally had positive comments about the paper, highlighting the quality of the writing, the rigor of the empirical study, and the impressive results.

**Weaknesses:**
Most reviewers felt that the technological novelty w.r.t. Diffusion-LM was minimal, though they acknowledged the differences. One reviewer had specific concerns about the justification for the need for a new type of diffusion specific to Seq2Seq problems. Another concern was that the authors had evaluated using non-standard tasks.

I’m inclined to accept the paper. I think the tasks in the empirical study are fine and may even provide promising indicators for applicability of the method beyond MT and summarization. With regards to whether the seq2seq-specific diffusion is necessary, the point was not deemed crucial by the reviewer who participated in the discussion, who felt the remaining contributions of the work made it suitable for acceptance.


**Note From Pc:**

if the above contains the word "oral" or "spotlight" please see: "oral" presentation means -> notable-top-5% and "spotlight" means -> notable-top-25%. As stated in our emails, we are disassociating presentation type from AC recommendations

**Summary Of Ac-Reviewer Meeting:**

Only one reviewer showed up. They mentioned they thought the paper made a quality contribution in an emerging research area and that they personally had learned a lot from reading it.